# Fluorescent secreted bacterial effectors reveal active intravacuolar proliferation of *Listeria monocytogenes* in epithelial cells

**Caroline Peron-Cane**[1,2], **José-Carlos Fernandez**[2], **Julien Leblanc**[2], **Laure Wingertsmann**[2], **Arnaud Gautier**[3,4], **Nicolas Desprat**[1,2,5]\*, **Alice Lebreton**[2,6]\*

**1** Laboratoire de Physique de l'École normale supérieure, ENS, Université PSL, CNRS, Sorbonne Université, Université de Paris, Paris, France, **2** Institut de biologie de l'ENS (IBENS), École normale supérieure, CNRS, INSERM, Université PSL, Paris, France, **3** Sorbonne Université, École normale supérieure, Université PSL, CNRS, Laboratoire des Biomolécules, LBM, Paris, France, **4** Institut Universitaire de France, **5** UFR de Physique, Université Paris-Diderot, Université de Paris, Paris, France, **6** INRAE, IBENS, Paris, France

\* nicolas.desprat@ens.psl.eu (ND); alice.lebreton@ens.psl.eu (AL)

**Data Availability Statement:** All relevant data are within the manuscript and its Supporting Information files.

## Abstract

Real-time imaging of bacterial virulence factor dynamics is hampered by the limited number of fluorescent tools suitable for tagging secreted effectors. Here, we demonstrated that the fluorogenic reporter FAST could be used to tag secreted proteins, and we implemented it to monitor infection dynamics in epithelial cells exposed to the human pathogen *Listeria monocytogenes* (*Lm*). By tracking individual FAST-labelled vacuoles after *Lm* internalisation into cells, we unveiled the heterogeneity of residence time inside entry vacuoles. Although half of the bacterial population escaped within 13 minutes after entry, 12% of bacteria remained entrapped over an hour inside long term vacuoles, and sometimes much longer, regardless of the secretion of the pore-forming toxin listeriolysin O (LLO). We imaged LLO-FAST in these long-term vacuoles, and showed that LLO enabled *Lm* to proliferate inside these compartments, reminiscent of what had been previously observed for Spacious *Listeria*-containing phagosomes (SLAPs). Unexpectedly, inside epithelial SLAP-like vacuoles (eSLAPs), *Lm* proliferated as fast as in the host cytosol. eSLAPs thus constitute an alternative replication niche in epithelial cells that might promote the colonization of host tissues.

## Author summary

Bacterial pathogens secrete virulence factors to subvert their hosts; however, monitoring bacterial secretion in real-time remains challenging. Here, we developed a convenient method that enabled fluorescent imaging of secreted proteins in live microscopy, and applied it to the human pathogen *Listeria monocytogenes*. *Listeria* has been described to invade cells and proliferate in their cytosol; it is first internalized inside vacuoles, from where it escapes thanks to the secretion of virulence factors that disrupt membranes. Our work revealed the existence, in human epithelial cells, of a population of *Listeria* that failed to escape vacuoles but instead multiplied efficiently therein, despite—and in fact, thanks

**Funding:** This project received support from ANR (LiVaLife-CE15-PRC-2020) for AL's and ND's groups. Work in the group of AL also received support under the program "Investissements d'Avenir" implemented by ANR (ANR-10-LABX-54 MemoLife and ANR-10-IDEX-0001-02 PSL University), Fondation pour la Recherche Médicale (FRM-AJE20131128944), Inserm ATIP-Avenir and Mairie de Paris (programme Émergences – Recherche médicale). The group of ND contributes to the IdEx Université; de Paris (ANR-18-IDEX-0001) and is part of "Institut Pierre-Gilles de Gennes" ("Investissements d'Avenir" program ANR-10-IDEX-0001-02 PSL and ANR-10-LABX-31) and the Qlife Institute of Convergence (PSL). CPC received a doctoral fellowship from programme Interface pour le Vivant from Sorbonne University. The funders had no role in study design, data collection and analysis, decision to publish, or preparation of the manuscript.

**Competing interests:** I have read the journal's policy and the authors of this manuscript have the following competing interests: AG is co-founder and holds equity in Twinkle Bioscience/The Twinkle Factory, a company commercializing the FAST technology. FAST was patented by AG and L. Jullien (Patent Publication# WO/2016/001437, International Application# PCT/EP2015/065267).

to—the active secretion of a toxin that permeates membranes. This intravacuolar niche may provide *Listeria* with an alternative strategy to colonize its host.

## Introduction

Bacterial pathogens harness distinct colonization strategies to take advantage of their host resources. While some remain extracellular, others adopt an intracellular lifestyle. Internalisation into host cells provides invasive bacteria with multiple abilities, such as that of crossing cellular barriers, escaping humoral immune surveillance, or disseminating throughout the organism as cargo of circulating cells. After internalisation, bacteria are entrapped inside primary vacuoles from where they can follow two distinct routes: either subverting endomembrane compartments, or leaving them. For instance *Chlamydia trachomatis*, *Brucella abortus* or *Legionella pneumophila* perturb the maturation and rearrange the properties of vacuoles, thereby creating a compartment prone to their replication [1]. Others, such as *Shigella flexneri* or *Listeria monocytogenes*, typically do not grow inside endomembrane compartments, but rather escape from entry vacuoles and gain access to the host cell cytoplasm, where they can replicate as well as exploit the host actin cytoskeleton for intracellular motility and cell-to-cell spread [2].

The foodborne pathogen *Listeria monocytogenes* (hereafter, *Lm*) is the causative agent of listeriosis, and has emerged as a model facultative intracellular bacterium [3,4]. This pathogen can cross the protective barriers of its host and colonize tissues and organs by promoting its internalisation into non-phagocytic cells. The classical scheme of *Lm* intracellular life cycle implies that, both in professional phagocytes and in epithelial cells, *Lm* rapidly escapes from entry vacuoles due to the combined action of a potent pore-forming toxin, listeriolysin O (LLO), and of two phospholipases C (PlcA and PlcB), before replicating in the cytosol [5]. All three genes (*hlyA* that encodes LLO, *plcA* and *plcB*) are part of *Lm* LIPI-I virulence gene cluster and are transcriptionally induced by PrfA, the main regulator of *Lm* virulence gene, in intracellular bacteria [6].

LLO is a cholesterol-dependent pore-forming toxin secreted by *Lm* via the general secretion system (Sec) [7]. LLO assembles into oligomers on biological membranes, forming arcs and pores of several tens of nm that disrupt membrane integrity [8]. Its activity is optimal at acidic pH representative of the acidification occurring during the maturation of phagosomes (pH = 4.9 to 6.7), which has been proposed to facilitate the escape of bacteria from entry vacuoles while avoiding damages to the host plasma membranes at neutral pH [9,10]. Whereas LLO-deficient *Lm* cannot gain access to the host cytosol in many cell types, the activity of the phospholipases PlcA and PlcB and the influence of host factors render LLO dispensable for vacuole escape in several human epithelial cell lines [11]. In phagocytes, it has been shown that bacteria secreting reduced amounts of LLO could remain entrapped in long-term compartments named Spacious *Listeria*-containing Phagosomes (SLAPs) and even replicate extremely slowly therein, with a doubling time in the range of 8 h [12].

The escape dynamics from the entry vacuole has been experimentally addressed using several distinct strategies. One of them consisted in using medium containing a membrane-impermeant fluorescent dye during infection [9]. Upon encapsulation into the internalisation vacuoles together with invading bacteria, the fluorescent dye stained the intravacuolar space until it broke down. However, this method required a washing step after bacterial entry in order to remove the unwanted background due to the high extracellular amount of the dye. This washing step thus prevented the observation of the first 10 minutes of the infection

dynamics. Alternative strategies were based on the assessment of vacuole rupture events and bacterial access to the host cytosol using fluorescent sensors. For instance, galectin-3 was shown to label membrane remnants of damaged vacuoles and thereby allowed the spotting of vacuole lysis [13]. Likewise, actin or the Cell-wall Binding Domain CBD (a domain from the *Lm* phage endolysin Ply118) are recruited to the bacterial surface only once *Lm* has escaped the vacuole [5,14]. Cytoplasmic FRET probes that are cleaved by a β-lactamase secreted by invasive bacteria have also been described as efficient reporters of average vacuole rupture time at a cellular scale [15,16]. Although these approaches yielded the order of magnitude of the time lapse between bacterial entry and vacuole escape in various cell types (between 15 min and 1 h), they did not allow a precise recording of the onset of entry events, which prevented their use for establishing the distribution of *Lm* residence time in entry vacuoles with accuracy. These constraints have limited the possibilities of refined quantitative comparisons of variations in intravacuolar residence times between different conditions.

In order to measure the heterogeneity of *Lm* residence time in entry vacuoles and to assess the role played by LLO in the dynamics of bacterial escape from these compartments, we developed live imaging assays allowing an accurate measurement of the time elapsed between the moment when individual bacteria were internalised into cells and the moment when the integrity of the vacuole membrane was disrupted. We devised a strategy based on tagging proteins secreted by bacteria with the FAST reporter system [17]. FAST is a 14-kDa protein tag which displays fluorescence upon binding with a synthetic fluorogenic probe supplied in the medium. The fluorogen is membrane permeant, non-toxic, and has very little fluorescence by itself. The small size of FAST, its rapid folding kinetics, the reversible binding of fluorogens together with good brightness and photostability made this system an ideal candidate for tagging secreted proteins such as LLO and imaging them in real time.

Using live imaging of FAST-tagged proteins, we quantified the distribution of *Lm* residence times in primary vacuoles in the LoVo intestinal epithelial cell line. We observed that a fraction of the population of entry vacuoles lasted for several hours and were reminiscent of SLAPs. However, in contrast with SLAPs, the prolonged residence of *Lm* inside vacuoles was observed in cells infected with wild type (WT) *Lm* as well as with a *hlyA* deletion strain. The secretion of LLO allowed *Lm* to proliferate actively inside these compartments, suggesting that besides its role in vacuole escape, LLO may contribute to setting up an intravacuolar niche permissive for *Lm* replication in epithelial cells.

## Results

### Fluorescent tagging with FAST of proteins secreted by *Listeria monocytogenes*

With the aim of detecting proteins that were secreted by intracellular bacteria into their host cells in live-cell microscopy experiments, we explored the possibilities offered by the FAST reporter system for the fluorescent tagging of *Lm* secreted bacterial effectors. A set of integrative plasmids harbouring gene fusions under control of the $P_{HYPER}$ promoter were designed (Fig 1A) and introduced in the genome of *Lm* strain LL195. These plasmids drove the constitutive production of either FAST or eGFP, either for intrabacterial localisation, or fused in their N-terminus to the secretion signal peptide (SP) of listeriolysin O (LLO) (SP-FAST and SP-eGFP constructs), or to full-length LLO (LLO-FAST, LLO-eGFP and untagged LLO constructs), a classical Sec substrate. A Myc tag in the C-terminus of all constructs allowed detection by immunoblotting. Protein production and secretion by each one of these seven strains was assessed by in-gel colloidal Coomassie staining and immunoblotting against the Myc tag, on bacterial total extracts and culture supernatant fractions from 16-h cultures in brain heart

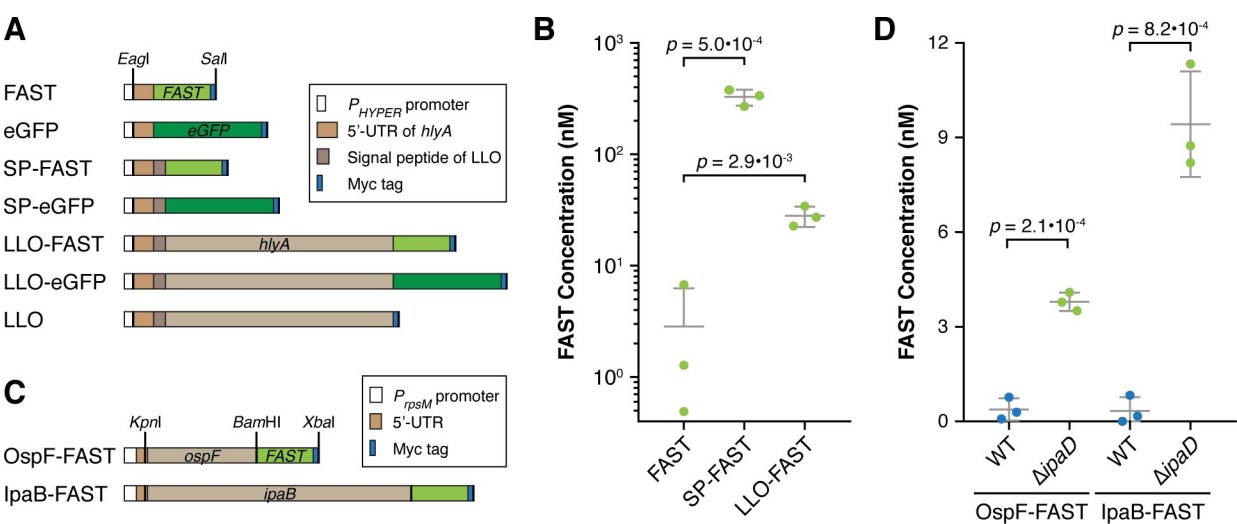

**Fig 1. FAST-tagged proteins retain fluorescent properties after secretion into bacterial culture media.** (A) Diagram of constructs in the pAD vector for constitutive expression in *Lm*. (B) *Lm* strains expressing FAST-tagged proteins were cultured in LSM, then fluorescence intensities were measured on the filtered supernatants of each culture in presence of 5 μM HBR-3,5DM. Concentrations of FAST-labelled proteins were calculated by reference to a standard curve of purified FAST in LSM. Residual fluorescence measured for the strain producing non-secreted FAST represents bacterial lysis. (C) Diagram of constructs in the pSU2.1 vector for expression in *Sf*. (D) WT or Δ*ipaD Sf* strains expressing FAST-tagged OspF or IpaB were cultured in M9 medium, then fluorescence intensities were measured on the filtered supernatants of each culture in presence of 5 μM HBR-3,5DM. Concentrations of FAST-labelled proteins were calculated by reference to a standard curve of purified FAST in M9 medium. Whereas Δ*ipaD* strains secrete proteins constitutively, T3SS secretion is not activated in WT strains, thus the fluorescent signals measured for these strains (blue dots) reflect bacterial lysis and/or leakage of the T3SS. (B, D) Normalized values, means and standard deviations from three independent experiments were plotted. *p*-values represent the results of two-tailed Student's *t*-tests with equal variance assumption. Source data are provided in S3 Table.

infusion (BHI) (S1 Fig). All transgenes were efficiently expressed by *Lm*, even though in varying amounts. As expected, constructs harbouring either the LLO SP, or full-length LLO, were recovered in bacterial culture supernatants, indicating that the SP of LLO promoted Sec-dependent export of FAST or FAST-tagged proteins, as well as eGFP-fusion proteins albeit to a lesser extent (S1C and S1D Fig). Constructs devoid of signal peptides were not detected in supernatant fractions, arguing against the release of proteins into the medium due to bacterial lysis. FAST-tagged Sec substrates can thus efficiently undergo secretion through the general secretion pathway.

To assess whether the FAST reporter system remained fluorescent after secretion, we quantified the fluorescence signals in the filtered culture medium of bacteria grown for 6 h in *Listeria* synthetic medium (LSM) (Fig 1B). In presence of 5 μM of HBR-3,5DM, fluorescence was detected in the culture supernatant of strains secreting SP-FAST or LLO-FAST. By calibrating fluorescence measurements with a standard curve of known FAST concentrations diluted in the same medium, we estimated the concentration of secreted tagged proteins; that of SP-FAST reached 325 ± 55 nM, and that of LLO-FAST was 28 ± 6 nM. In contrast, fluorescence levels in the culture medium of strains producing non-secreted FAST remained low, indicating that the release of fluorescent proteins in the culture medium due to bacterial lysis was minor. We conclude that FAST-labelled proteins retained their fluorescent properties after undergoing secretion through Sec.

Diverse attempts by others in Gram–negative bacteria [18] and our own observations using tagged *Lm* virulence factors suggested that the Sec-dependent secretion and subsequent maturation of an eGFP tag into its active, fluorescent fold was inefficient. Surprisingly, the secretion of SP-eGFP—but not that of LLO-eGFP—also gave rise to fluorescent signals in culture supernatants, even though in a range 10-fold lower than that obtained for the secretion of SP-FAST

(S2 Fig). A consistent proportion of eGFP undergoing Sec-dependent secretion was thus able to acquire its mature fold in bacterial culture medium, at least in conditions where it was not fused to a bacterial effector and in LSM.

## Fluorescent tagging with FAST of *Shigella* effectors secreted through the type III secretion system

To evaluate the versatility of FAST as a reporter of bacterial secretion, we next asked if FAST was suitable for fluorescent tagging of effectors secreted through the syringe of the type III secretion system (T3SS) from a Gram-negative pathogen, *Shigella flexneri* (*Sf*) strain M90T. As model T3SS substrates, we tagged the C-terminal ends of the effectors OspF and IpaB with FAST-Myc (Fig 1C), which are translocated upon adhesion of *Sf* to host cells [19]. Bacterial total extracts and culture supernatant fractions were recovered from 16-h cultures in M9 medium, with or without stimulation of type-III dependent secretion by addition of Congo red. By immunoblotting these fractions against the Myc epitope, we observed that tagged OspF and IpaB were secreted into the bacterial culture medium upon Congo red induction (S3A Fig). The secretion of both tagged effectors was constitutive when using a Δ*ipaD* mutant strain for which translocation lacks gating controls [20] (S3B Fig). We then assessed whether the fusion proteins secreted by the Δ*ipaD* strain had retained their fluorescent properties, by measuring fluorescence intensities in the supernatants of 16-h bacterial cultures in M9 medium (Fig 1D). Fluorescence levels were consistently higher with this constitutively secreting strain Δ*ipaD* than the fluorescence leakage measured for the WT strain when the T3SS was not induced. The concentration of OspF-FAST by the Δ*ipaD* strain was estimated to be 3.8 ± 0.3 nM, that of IpaB-FAST of 9.4 ± 1.7 nM. Like Sec substrates, FAST-tagged T3SS substrates can thus pass through the needle of the TS33, and keep fluorescent properties after secretion at least when gating controls are lacking.

## FAST-tagging of secreted *Listeria* effectors for live fluorescence microscopy

We next investigated whether the FAST reporter system was suited for detecting proteins secreted into the cytoplasm of host cells by real-time microscopy during infection. To this end, we monitored FAST signals in LoVo cells infected with *Lm* producing SP-FAST by confocal spinning disk microscopy over an infection time course (Fig 2 and S1 Movie). FAST fluorescence labelled uniformly the cytoplasm of infected cells and increased over time (Fig 2A). At 562 nm (the emission wavelength specific for FAST:HBR-3,5DM), fluorescent signals accumulated in cells infected with a strain producing SP-FAST, and not with a control isogenic strain that constitutively expressed mCherry (Fig 2B). In infected cells, measured fluorescence intensities—which reflects the intracellular concentration of SP-FAST—increased exponentially with time (Fig 2C), likely mirroring the exponential growth of *Lm* in the host cytosol. After several hours of signal accumulation, the intracellular fluorescence dropped abruptly. This corresponded to a sudden shrinkage of infected cells, probably resulting from their death and from the concomitant permeation of their membranes. For each cell, we fitted the dynamics of fluorescent signals to an exponential curve as shown in Fig 2C (black curve), measured the rates of fluorescence increase *r* for each exponential curve, calculated fluorescence doubling times ($\tau = \frac{ln2}{r}$), and then plotted their distribution (Fig 2D). The mean doubling time of FAST signals was 106.7 ± 41.9 min (n = 39). This value, which represents the characteristic time for SP-FAST accumulation in infected cells, was comparable to the mean doubling time of bacteria (94.7 ± 7.9 min, n = 4) measured for mCherry-labelled bacteria in similar conditions of infection and illumination (S4 Fig). Altogether, the secretion of FAST into host cells allowed a quantitative monitoring of infection progression by live imaging of individual cells.

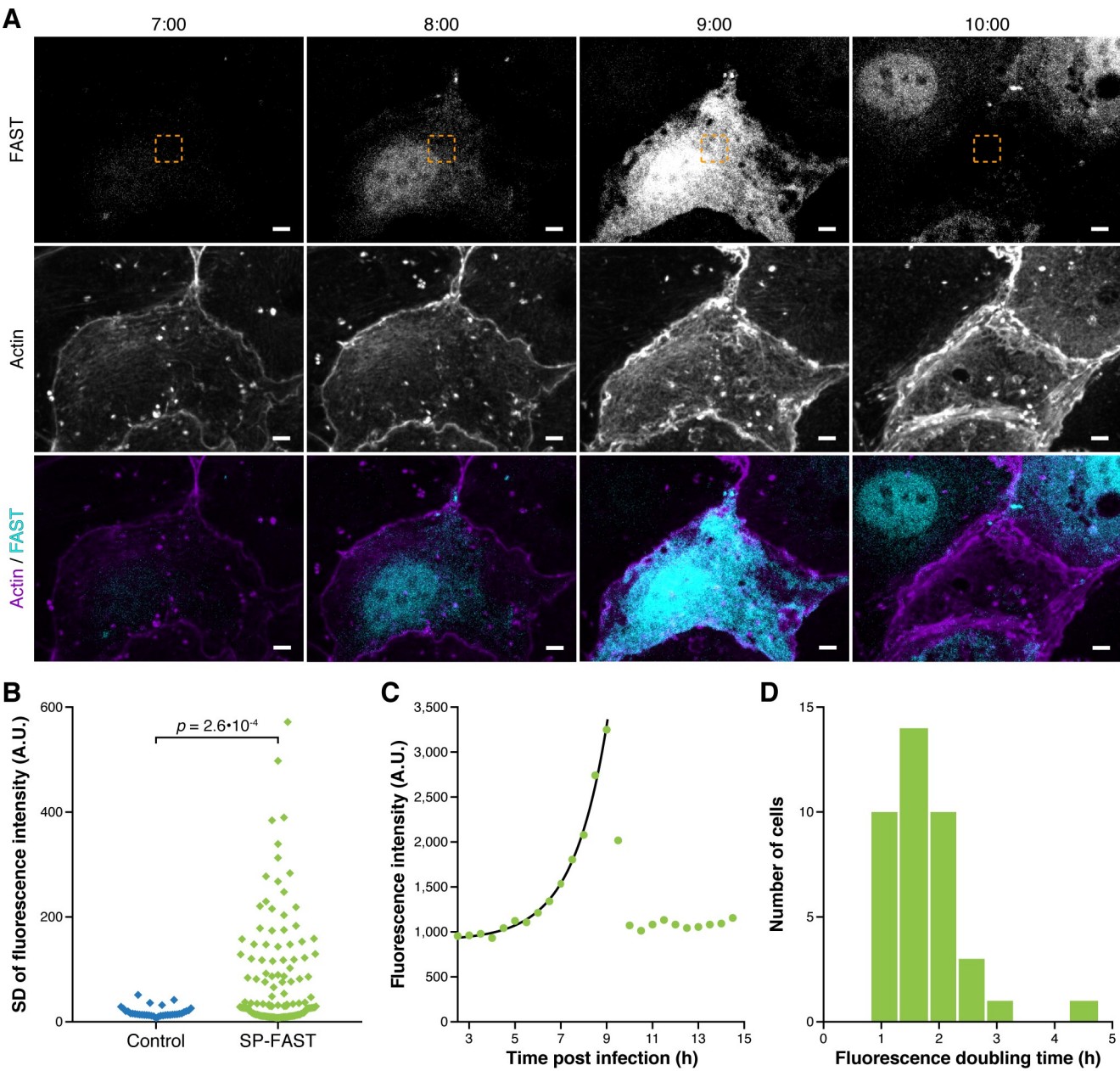

**Fig 2. Secreted FAST accumulates exponentially in the cytoplasm of infected cells.** (A) Spinning disk fluorescence microscopy images of LoVo cells infected with *Lm* secreting SP-FAST (cyan) at different time-points post-infection (h:min). The actin cytoskeleton (purple) was labelled with SiR-actin. The area where FAST fluorescence intensity was measured for graph (C) is boxed in orange. Scale bars, 5 μm. (B) Dispersion of fluorescence intensities. Fluorescence emission at 562 nm (FAST:HBR-3,5DM channel) was quantified over time within a region of fixed area in cells infected by *Lm* strains expressing either SP-FAST (in green, n = 127) or mCherry as a negative control (in blue, n = 35). As an indicator of the amplitude of fluorescence accumulation, the standard deviation of fluorescence intensity over time was plotted for each cell. A.U., arbitrary units. The *p*-value represents the result of a two-tailed Mann-Whitney non-parametric test. (C) Intensity of FAST signals measured over time in the region boxed in yellow in (A). The black line displays an exponential fit obtained over the ascending part of the curve (green dots). (D) Distribution of the doubling time of FAST fluorescence signals among the population of infected cells (n = 39). Source data are provided in S4 Table.

## Residence time of *Listeria monocytogenes* in internalisation vacuoles

When FAST-tagged proteins were secreted into the large volume of the host cell cytoplasm, fluorescent signals were diluted and therefore only became clearly visible after several hours of

infection, once secreted FAST had accumulated sufficiently to be significantly discriminated from non-specific signals. Meanwhile, we reasoned that if *Lm* was confined inside micron-sized internalisation vacuoles, the higher concentration of secreted FAST molecules in a reduced volume would allow their detection and tracking until the disruption of vacuole membranes, thereby providing an accurate measurement of individual vacuole lifetimes (Fig 3A). Indeed, we observed that secreted FAST signals were enhanced in compartments that co-localized with mCherry-expressing bacteria within minutes after bacterial adhesion to cells, until these signals suddenly dropped, most likely when vacuoles ruptured (Fig 3B and S2 Movie).

We used SP-FAST secretion to track intravacuolar fluorescent signals and compare the residence time of WT or Δ*hlyA Lm* strains inside internalisation vacuoles formed in LoVo cells (Fig 3C). The *hlyA* deletion strain used in this experiment was generated by in-frame allelic replacement of the *hlyA* open reading frame with SP-FAST (Δ*hlyA*::*SP-FAST*, S5A Fig) and also expressed mCherry constitutively. The growth rates of these two strains in BHI were undistinguishable (S5B Fig). The median value for the residence time of the WT strain was 12.7 ± 0.7 min (Fig 3C). When using the Δ*hlyA*::*SP-FAST* strain, the distribution of residence times was significantly shifted compared to the WT ($p$ = 0.0191). The median residence time was longer (21.1 ± 1.4 min) but remained of the same order of magnitude as for a strain producing LLO, confirming previous observations that *Lm* gained efficient access to the cytoplasm independently of LLO in epithelial cells [11]. Unexpectedly, a consistent proportion of the entry vacuoles lasted for more than one hour (12.0% for the WT strain; 14.8% for the Δ*hlyA* mutant), and intact vacuoles were still observed 3 h p.i. (4.6% for the WT strain; 6.2% for the Δ*hlyA* mutant) (Fig 3C). The fact that the WT strain remained entrapped in vacuoles in proportions nearly identical to that of the Δ*hlyA* strain could either suggest that a sub-population of WT *Lm* failed to escape primary vacuoles in spite of LLO secretion, or that LLO was not produced by this sub-population of intravacuolar bacteria. To discriminate between these two hypotheses, we investigated whether LLO fused to a FAST tag was detected in vacuoles from which *Lm* had failed to escape.

## Long-term residence and rapid replication of *Listeria* inside LLO-decorated vacuoles

To examine whether LLO was produced and secreted by bacteria that remained entrapped in vacuoles, we engineered a *Lm* strain where LLO was C-terminally fused with FAST at the endogenous *hlyA* locus (S5A Fig). In this strain, the fluorescence of FAST reported not only for LLO secretion and localisation, but also for *hlyA* expression under its natural promoter. In order to be relevant for monitoring the dynamics of *Lm* intracellular infection, the 15-kDa FAST-Myc tag should not interfere with the function of the protein it reports for. We controlled that the haemolytic properties of the strain expressing *hlyA-FAST* did not differ from that of the WT strain (S5C Fig); the production, secretion and activity as a cytolysin of LLO are thus quantitatively and qualitatively preserved after C-terminal fusion with FAST.

The strain producing the LLO-FAST fusion also constitutively expressed mCherry, which allowed us to segment and track bacteria in 3D during infection. When imaging mCherry-labelled bacteria and LLO-FAST from 2 h post-infection (p.i.) in LoVo cells, we observed that *Lm* could remain entrapped inside vacuoles for several hours before the enclosed structure of LLO-labelled membranes eventually disrupted and bacteria dispersed into the cytosol (Fig 4A and S3 Movie). On Fig 4A, the two vacuoles indicated with arrowheads ruptured after 4 h 25 min and 7 h of infection, respectively. A number of these vacuoles lasted even longer, and up to 9 h p.i., when observations were interrupted. Strikingly, the size of these compartments and the number of mCherry-labelled bacteria they contained increased over time, revealing that some

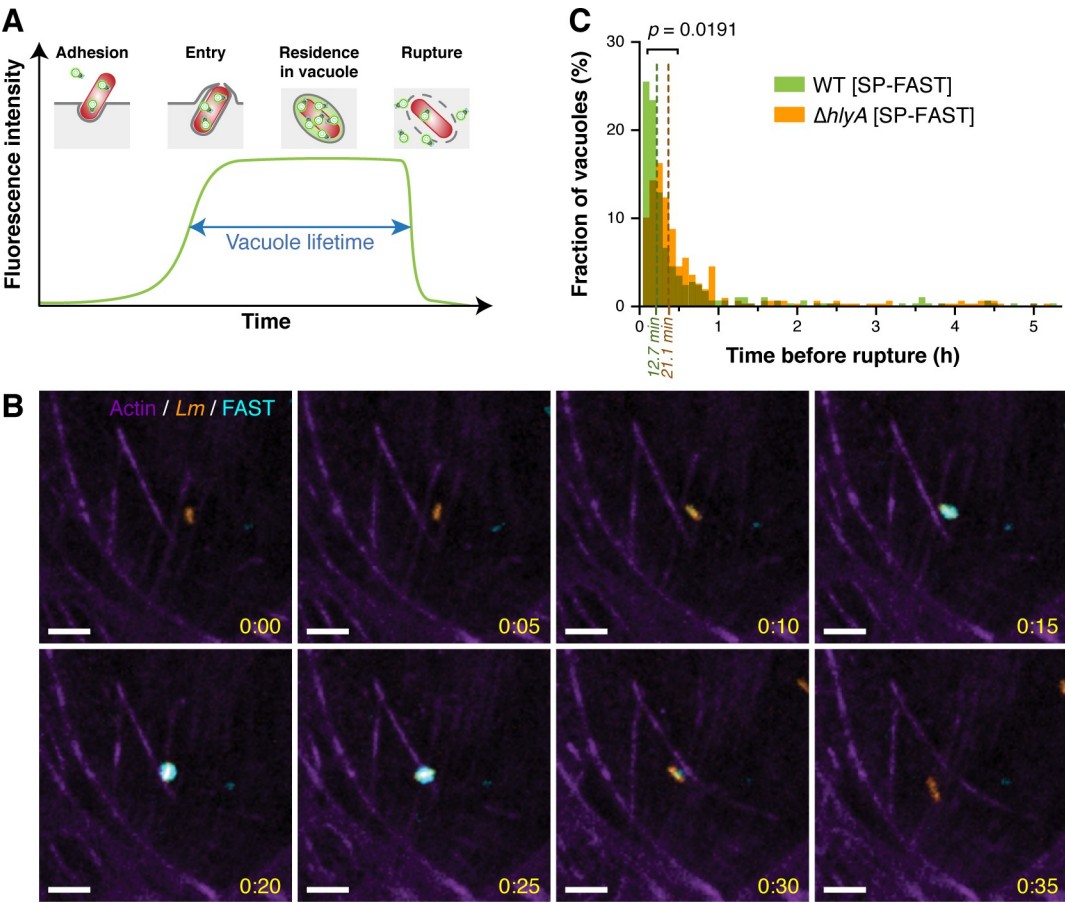

**Fig 3. Secreted FAST reveals the heterogeneity of *Listeria* residence time in internalisation vacuoles.** (A) Expected profile of fluorescence accumulation in internalisation vacuoles for *Lm* secreting SP-FAST. After bacterial adhesion, *Lm* enters epithelial cells via a zipper mechanism. Secreted FAST should start accumulating in vacuoles upon their closure, and then remain visible until vacuole rupture. In each vacuole, the level of fluorescence reflects the equilibrium between bacterial secretion of FAST and its leakage in case of membrane permeation. (B) Spinning disk microscopy images of LoVo cells infected with *Lm* Δ*hlyA* expressing SP-FAST (cyan) and mCherry (orange) for 35 min after entry. The actin cytoskeleton (purple) was labelled with SiR-actin. Scale bars, 5 μm; timescale, h:min. (C) Distribution of *Lm* residence times in internalisation vacuoles in LoVo cells. Green, WT strain carrying an integrated pAD-*SP-FAST* plasmid (n = 284); orange, Δ*hlyA*::*SP-FAST* strain carrying an integrated pHpPL3-*mCherry* plasmid (n = 306). The interpolated median lifetime of SP-FAST-labelled vacuoles, calculated from the raw distributions, are displayed in dark green and dark orange dashed lines for the WT and Δ*hlyA* strains, respectively. The *p*-value indicates the result of a two-tailed Student's *t*-test on the distributions, assuming equal variance. Source data are provided in S5 Table.

bacteria not only inhabited vacuoles for a long time, but also efficiently multiplied therein. The ability of *Lm* to grow inside LLO-FAST-labelled vacuoles was observed for both the LL195 genetic background (*Lm* lineage I, ST1) (Fig 4A and S3 Movie) and the European *Lm* reference strain EGD-e (lineage II, ST9) (S6A Fig and S4 Movie), indicating that this property was not specific to the hypervirulent clone LL195. Likewise, the proliferation of *Lm* inside long-term vacuoles decorated with LLO-FAST was observed in Caco-2 cells, suggesting that LoVo cells were not the only epithelial niche allowing *Lm* to replicate inside endomembrane compartments (S6B Fig). Because these vacuoles were reminiscent of the SLAPs previously described in macrophages [12], we will refer to them as eSLAPs (for epithelial SLAP-like vacuoles) hereafter.

By tracking vacuoles and segmenting the mCherry fluorescence signals, we computed the volume occupied by intravacuolar bacteria, which is proportional to their number, and thereby

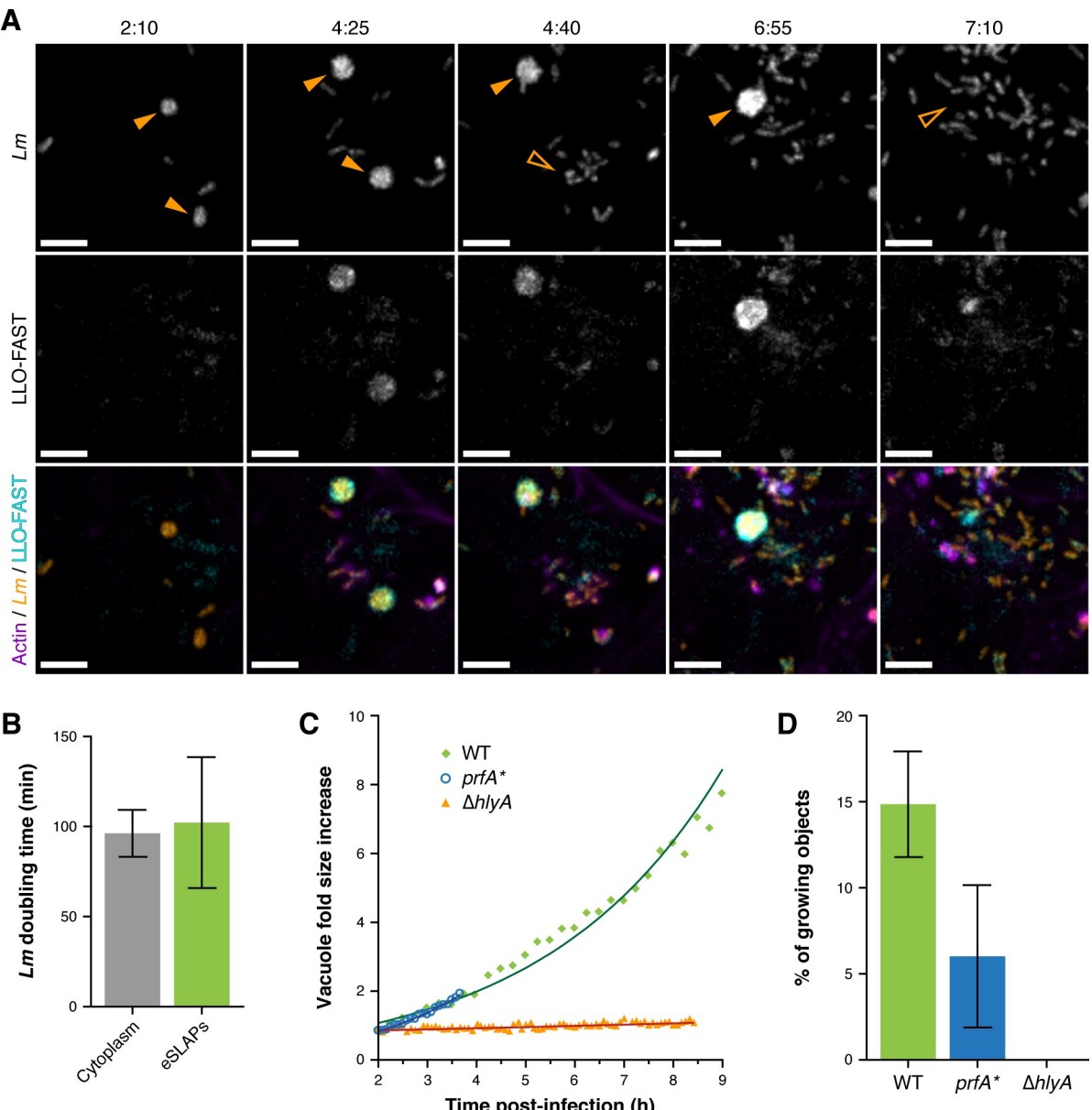

**Fig 4. *Listeria monocytogenes* replicates inside long-term vacuoles decorated with LLO.** (A) Spinning disk microscopy images of LoVo cells infected with *Lm* expressing both LLO-FAST (in cyan) and mCherry (in orange) at several time-points post-infection. SiR-actin staining is shown in purple. eSLAPs are indicated with solid orange arrowheads; their past location is pointed with open arrowheads after their rupture. Scale bars, 5 μm; timescale, h:min. (B) Doubling times of *Lm* expressing mCherry in the cytoplasm (grey, n = 7) or in eSLAPs (green, n = 18) in infected LoVo cells. (C) Quantification of the increase in volume of eSLAPs, and thus of the growth of the bacteria they contain, for WT (green), *prfA** (blue) or Δ*hlyA* (orange) *Lm* strains in infected LoVo cells. (D) Proportion of intracellular *Lm* that multiplied inside eSLAPs during a time-course of 8 h. Plotted values represent the ratio of the number of eSLAPs that had at least doubled in volume over the time course to the number of segmented mCherry objects (*i.e.* cytoplasmic or intravacuolar bacteria, isolated or in clusters) at the beginning of the observation (2 h p.i). Green, WT strain (n = 134); blue, *prfA** strain (n = 33); orange (null), Δ*hlyA* strain (n = 113). Source data are provided in S6 Table.

determined the growth rate of WT *Lm* in eSLAPs (S7A Fig). Segmenting mCherry-labelled bacteria also allowed the measurement of growth rates of free bacteria in the cytosol (S8 Fig). Like cytosolic growth, intravacuolar growth was exponential, with a mean doubling time (102.2 ± 36.3, n = 18) similar to that of cytosolic bacteria (96.3 ± 13.0, n = 7) (Fig 4B). This

doubling time was consistently faster than that previously described in SLAPS, which was in the range of 8 h [12].

## Role of listeriolysin O in the long-term intravacuolar residence and replication of *Listeria*

Our above-mentioned results showed that LLO-FAST was secreted by *Lm* and functional, and also that it was present in eSLAPs, which is likely to permeate their membranes. However, despite the presence of LLO, the integrity of eSLAPs was preserved over several hours, and intravacuolar replication of *Lm* occurred without vacuole rupture. To determine whether LLO concentration influenced *Lm* residence time in eSLAPs, we took advantage of the LLO-FAST reporter strain in order to measure the variability in LLO abundance in vacuoles. LLO-FAST signals measured in eSLAPs displayed a broad spectrum of dynamics, indicating that eSLAP formation and duration were independent of the amounts of secreted LLO (S7B Fig). In some vacuoles, LLO-FAST accumulated linearly over time, while others displayed large-scale fluctuations in signals that may reflect variations in LLO synthesis, secretion, degradation, leakage from vacuole or membrane repair. Some eSLAPs yielded a strong signal while others displayed low levels of decoration by LLO. The lifetime of eSLAPs was correlated with neither the average nor the maximal level of LLO-FAST concentration in eSLAPs (S7C and S7D Fig), suggesting that LLO concentration had a limited influence on the probability of *Lm* escape from these structures.

To further assess the effects of LLO concentration on the stability of eSLAPS, we tracked intravacuolar bacteria for WT, Δ*hlyA* and *prfA** mutant strains (Figs 4C and S9). When using the *hlyA* deletion strain, lasting vacuoles were observed (Figs 3C, 4C and S9). However, Δ*hlyA* bacteria were unable to proliferate inside these long-lived vacuoles (Figs 4C and S9). Similar to SLAPs, eSLAPs thus required that bacteria secreted LLO to allow intravacuolar growth.

To examine whether the activity of LLO as a pore-forming toxin was required for bacterial growth in eSLAPs, we carried out the same experiment using a strain with a W492A point mutation in LLO that had been previously described to almost abolish haemolytic activity [21]. The introduction of this mutation in chromosomal *hlyA*-FAST in the LL195 background resulted in a strain that retained ~1 to 3% of the haemolytic titre of the isogenic WT strain (S5C Fig). These bacteria were still able to grow in vacuoles (S9 Fig), suggesting that either the pore-forming toxin activity of LLO was itself dispensable for allowing intravacuolar growth, or the residual activity of the toxin was enough to support growth.

Conversely, to assess the effects of increased LLO production, we used a *prfA** mutant strain to investigate the outcome of LLO overexpression. The *prfA** allele encodes a PrfA variant with a G145S substitution that has been previously described to be constitutively active, and to lead to the strong overexpression of PrfA-dependent virulence genes, including that of *hlyA* [22]. Accordingly, the *in-vitro* haemolytic titre of the LL195 *prfA** strain was fifty-fold higher than that of the isogenic WT strain, indicative of LLO hyperproduction (S5C Fig). Using WT or *prfA** reporter strains where eGFP was inserted by allelic replacement under control of the endogenous $P_{hlyA}$ promoter, we measured that eGFP expression was on average 6.7-fold higher in the *prfA** strain than in the WT strain at 1 h p.i., and 2.7-fold higher at 3 h p.i. (S10 Fig). In our experimental model, the *prfA** mutation thus also led to overexpression of *hlyA* by intracellular bacteria, at least in the first hours of infection when eSLAPs start being formed by intravacuolar bacteria. Despite higher levels of *hlyA* expression, eSLAPs were still detectable for the *prfA** strain, indicating that increased secretion of the pore-forming toxin did not hamper the ability of *Lm* to reside and multiply inside vacuoles during several hours. This feature contrasts with SLAPs, which formed in phagocytes only when the expression of *hlyA* was

moderate [12]. LLO quantity did not influence intravacuolar bacterial growth, since the *prfA** strain replicated at a similar rate as the WT strain in eSLAPs (Fig 4C). Consistently, the growth rate of LLO-FAST-secreting bacteria in eSLAPs was correlated with neither the average nor the maximal level of LLO-FAST fluorescence intensity (S7E–S7F Fig). Nevertheless, we observed by live-cell imaging that the escape of the *prfA** strain from eSLAPs occurred earlier than for the WT strain (Fig 4C and S9). In agreement with this observation, the ratio of eSLAPs to the initial number of entry events was lower when cells were infected with the *prfA** strain than with the WT strain (Fig 4D). This higher probability of vacuole escape for the *prfA** strain suggests that a high concentration of LLO exerts a mild destabilising effect on the integrity and duration of eSLAPs, though it does not preclude their formation.

Altogether, our results suggest that the secretion of LLO is required for proliferation of *Lm* in eSLAPs, but that its concentration and overall activity exert only a minor influence upon eSLAP lifetime and on the ability of bacteria to grow inside.

## Origin and properties of *Listeria* long residence vacuoles in epithelial cells

The eSLAPs in which *Lm* replicated (Fig 4 and S3 and S4 Movies) likely originated from internalisation vacuoles from which bacteria had failed to escape (Fig 3C), unless they derived from secondary vacuoles produced by cell-to-cell spread, or by autophagy vacuoles where bacteria would have been entrapped after a first exposure to the host cytoplasm. To assess whether eSLAPs resulted from primary vacuoles, we monitored the intravacuolar stages of mCherry-expressing bacteria in LoVo cells transfected with the YFP-CBD fusion protein reporter [14]. This reporter has been previously described to specifically label the surface of bacteria once exposed to the host cytosol, because the cell wall-binding domain (CBD) from the *Lm* phage endolysin Ply118 binds the peptidoglycan of *Lm* with high affinity. Bacteria that replicated within eSLAPs remained unlabelled with YFP-CBD until the vacuole ruptured and bacteria dispersed throughout the cell (Fig 5A and S5 Movie and S11A Fig), indicating that they had not been in prior contact with the host cytosol. This result ruled out the possibility that bacteria became entrapped into secondary vacuoles by canonical autophagy or cell-to-cell spread after a first exposure to the host cell cytosol, and thereby confirmed that eSLAPs where *Lm* replicated originated from internalisation vacuoles.

Because the replication compartments we observed were reminiscent of SLAPs, we hypothesized that they could originate from a process analogous to LC3-associated phagocytosis (LAP), except it would occur in epithelial cells rather than in phagocytes. We thus carried out a molecular characterization of this intravacuolar replication niche in order to analyse whether it had typical features of endosomal, lysosomal and/or noncanonical autophagy-derived compartments. By immunofluorescence staining of LoVo cells infected with mCherry-expressing *Lm* for 3 hours, we observed that the vacuoles containing several bacteria were negative for the early endosomal marker Rab5 (9% of colocalisation, n = 22), while they were positive for the late endosomal marker Rab7 (88.5%, n = 26), LC3 (100%, n = 63), as well as the lysosomal marker LAMP1 (80.5%, n = 41) (Figs 5B and S11B). These are typical markers of SLAPs, suggesting that, similar to what occurs in phagocytes, LC3 is lipidated and the noncanonical autophagy machinery recruited to entry vacuoles in epithelial cells. Also, as in SLAPs the pH inside eSLAPs remained neutral, which is revealed by their absence of staining when using the acidophilic fluorescent probe LysoTracker Deep Red (0%, n = 17) (Figs 5B and S11B). Altogether, we conclude that eSLAPs display molecular characteristics highly reminiscent of SLAPs, although they allow a faster replication of *Lm*, and their maturation and rupture is less sensitive to the concentration of secreted LLO than the compartments observed in phagocytes.

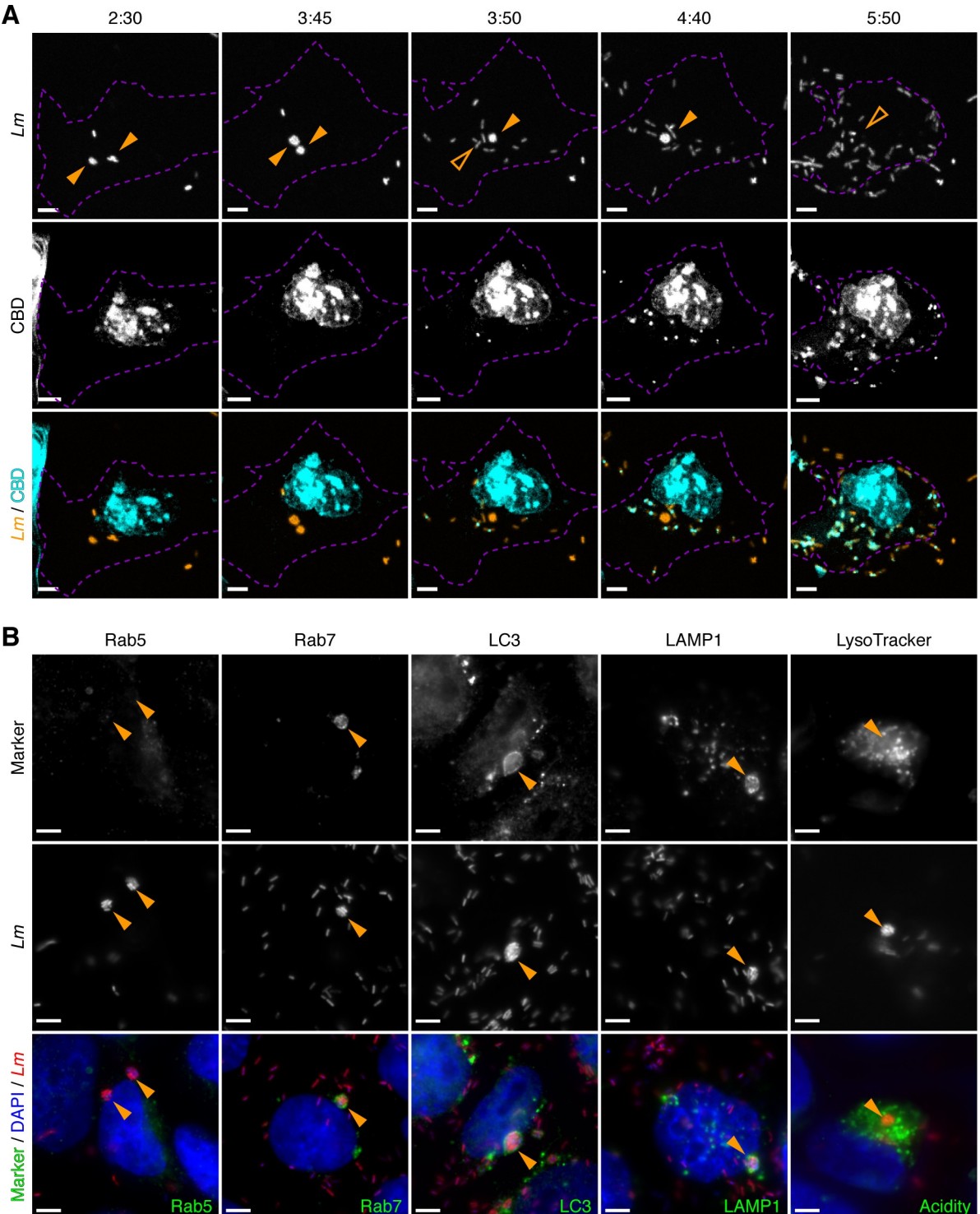

**Fig 5. *Listeria* eSLAPs derive from internalisation vacuoles and display typical markers of LC3-associated phagocytosis.** (A) Differential labelling by YFP-CBD of the cytosolic *versus* intravacuolar populations of intracellular bacteria. LoVo cells were transfected with pEYFP-CBD (in cyan) 24 h before being infected with *Lm* expressing mCherry (in orange), then imaged at different time-points post infection. eSLAPs are indicated with solid orange arrowheads; their past location is pointed with open arrowheads after their rupture. The cell outline is indicated with purple dashed lines. Note that a strong non-specific YFP-CBD signal is also detected in the cell nucleus. Timescale, h:min. (B) Rab5, Rab7, LC3 and LAMP1 (in green) were detected by immunofluorescence in LoVo cells infected for 3 h with mCherry-expressing bacteria (in red). For acidity staining, LoVo cells infected for 2 h with eGFP-expressing bacteria (in red) were stained

with LysoTracker Deep Red (in green), and observed 1 h afterwards on an inverted spinning disk microscope. Orange arrowheads point to representative eSLAPs. (A, B) Scale bars, 5 μm. Quantitative analyses of these experiments are provided as S11 Fig.

## Discussion

Exploring the dynamics of secreted virulence factors at the (sub-)cellular scale constitutes one of the main challenges for real-time microscopy of infectious processes. Here, we bring evidence that FAST offers a versatile, convenient opportunity for tackling this challenge. We took advantage of this system to measure the lifetime of *Lm* internalisation vacuoles, and to monitor the endomembrane localisation of the secreted *Lm* virulence factor LLO in live cells. As a result, we uncovered an intravacuolar replication niche for *Lm* in epithelial cells.

### Real-time imaging of LLO during infection

On fixed samples, observing the localisation of LLO in infected cells has often constituted a hurdle, due to the poor quality of the labelling allowed by existing anti-LLO antibodies in immunofluorescence assays [14]. LLO localisation at vacuole membranes, or more recently in bacterial-derived membrane vesicles, was first observed by electron microscopy using immunogold labelling [23,24]. However, the precise dynamics of infectious processes cannot accurately be caught by fixed-cell studies. Besides, the high spatial resolution gained by electron microscopy compromises the observation of events at a cellular scale. As a complementary approach, LLO-eGFP fusions that were ectopically-expressed in host cells have enabled live imaging, yielding precious insight into the dynamics of LLO localisation at membranes and its turnover [25]. Nevertheless, ectopic expression by host cells cannot mimic the concentrations, location, and insertion into membranes from the inside of the vacuole obtained with bacterial secretion. Moreover, host cell signalling pathways and membrane dynamics differ between non-infected and infected cells. Here, we report that (*a*) the FAST system can be used to tag LLO without loss of function, (*b*) the LLO-FAST fusion, expressed from its endogenous promoter, is secreted by *Lm* in infected cells, (*c*) the vacuoles it decorates can be imaged with accuracy, and (*d*) some of these vacuoles unexpectedly last for several hours.

### FAST, a versatile fluorescent reporter of bacterial secretion

Beyond the live detection of LLO secreted by *L. monocytogenes* through the general Sec secretion system, FAST opens new perspectives for real-time imaging of bacterial proteins secreted by a broader range of bacterial models and secretion systems. For instance, we provide data supporting that FAST-tagged effectors can also be efficiently secreted through the T3SS of *S. flexneri*.

In recent years, several strategies have emerged for fluorescent labelling of Sec–or T3SS–dependent substrates [26]. Tagging bacterial effectors with Split-GFP provides a possible solution that has been successfully applied for live detection of *Salmonella* T3SS-dependent effectors or *Listeria* Sec-dependent secreted substrates [27,28]; however, the reconstitution process is slow compared with microbial growth, and requires the stable expression of GFP$_{1-10}$ in recipient cells, which limits its application in most biological systems. Superfolder GFP (sfGFP) or its derivative rsFolder have been successfully used for labelling *E. coli* periplasmic proteins exported through the Sec pathway [18,29], but to our knowledge has not been applied yet for other bacterial systems or in the context of host-pathogen interactions. Other fluorescent tags such as FlAsH and phiLOV were successfully used for monitoring the secretion of *Sf* T3SS-dependent effectors [30,31]. Nevertheless, the toxicity in eukaryotic cells of the biarsenite dye used for FlAsH labelling and the rather modest brightness of phiLOV hamper their general use.

FAST compares with previously existing tools, while broadening the possible range of applications, due to (*a*) its ease of implementation (compared with Split-GFP); (*b*) its low toxicity (compared with FlAsH); (*c*) its independence to oxygen allowing studies in anaerobes [32,33] as well as (*d*) its rapid and reversible folding dynamics allowing transport through the T3SS (compared with GFP-derived probes); (*e*) its reasonable brightness and fast maturation time (compared with phiLOV). In addition, FAST-labelled proteins can be imaged at different wavelengths between 540 and 600 nm by selecting the appropriate fluorogen [34], thereby providing users with flexibility in the choice of other fluorescent reporters in co-localisation studies. Red-shifted fluorogens also limit the toxicity of certain wavelength for bacteria when performing long-term imaging, and membrane-impermeant fluorogens offer the possibility to discriminate between intracellular and extracellular proteins [35], for instance when addressing the localisation of bacterial effectors that anchor to the bacterial cell wall or to membranes [36].

Hence, FAST expands the panel of fluorescent reporters for monitoring secreted virulence factors and offers a wealth of opportunities to accurately seize the spatiotemporal aspects of infectious mechanisms.

## eSLAPs, an alternative replication niche for *Listeria monocytogenes* in epithelial cells

We document that in LoVo and Caco-2 epithelial cells, a consistent proportion of *Lm* fails to escape from internalisation vacuoles, but instead replicates efficiently inside epithelial SLAP-like vacuoles (eSLAPs), which are positively labelled by LLO-FAST (Fig 6). After several hours of intravacuolar residence and growth, eSLAPs eventually break open and bacteria resume a canonical cytosolic lifestyle.

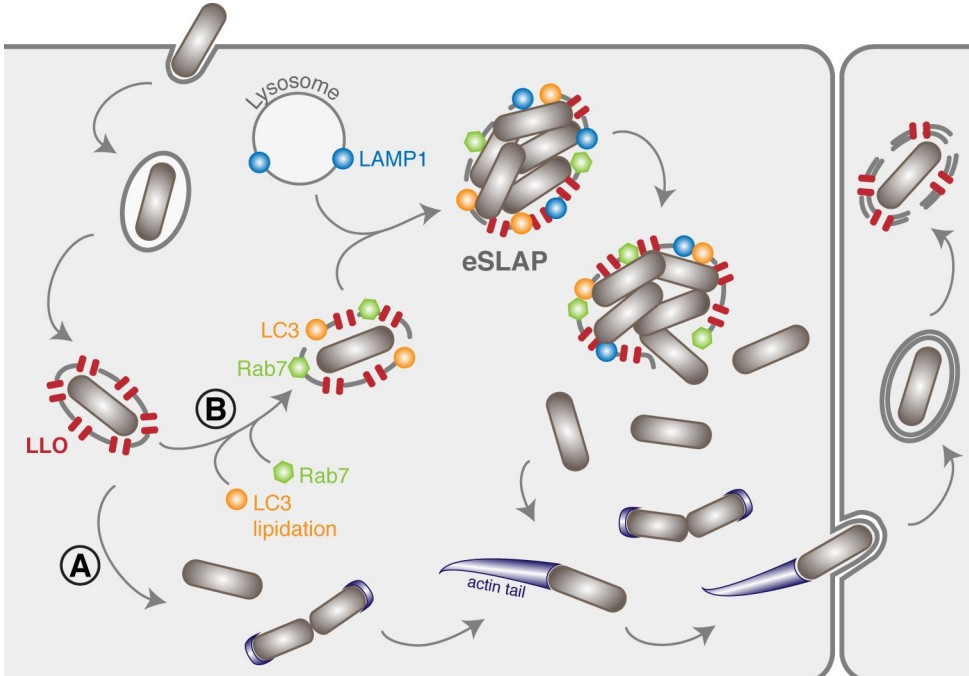

**Fig 6. Extended model of the intracellular life cycle of *Listeria monocytogenes* in colon adenocarcinoma epithelial cell lines.** (A) In the classical scenario, after receptor-mediated entry, *Lm* evades the vacuole thanks to the combined action of LLO and phospholipases. (B) Here we identified a population of *Lm* that can remain for several hours and multiply inside vacuoles in LoVo cells. These compartments (eSLAPs) are neutral, positive for Rab7, LC3 and LAMP1, and decorated with LLO. This second population of bacteria finally escapes into the cytoplasm at later time points.

The decoration of eSLAPs by LC3, Rab7 and LAMP1 as well as their neutral pH are reminiscent of the SLAPs (Spacious *Listeria*-containing Phagosomes) previously described in phagocytes [12], and which derive from LAP (LC3-associated phagocytosis) [37]. Upon infection by *Lm*, we propose a model for the formation of replicative eSLAPs, analogous to the current model of SLAP formation (Fig 6). The entrapment of *Lm* inside internalisation vacuoles could result in two distinct fates. (**A**) In the classically-described pathway, the coordinated actions of LLO, PlcA and PlcB result in a rapid disruption of the vacuole and escape of bacteria into the cytoplasm, where they can start replicating and polymerising actin. (**B**) In the second scenario, a proportion of internalisation vacuoles would undergo LC3 lipidation in addition to their maturation attested by decoration with Rab7, as well as fusion with lysosomes as suggested by LAMP1 labelling.

Whereas the eSLAPs observed in LoVo cells display similarities with SLAPs, they are notably distinct from LisCVs, which are an intravacuolar persistence niche of *Lm* recently described in human hepatocytes and trophoblast cells [38]. Contrary to SLAPs and eSLAPs, LisCVs do not derive from primary vacuoles. Instead, they form late in the intracellular cycle of *Lm* by recapture of bacteria that have lost ActA-dependent motility. Indeed, bacteria found in LisCVs are labelled with YFP-CBD, while the bacteria we observe in eSLAPs are not. Moreover, whereas eSLAPs are lipidated by LC3, LisCVs are not. Last, *Lm* replicates in eSLAPs, whereas it adopts a viable but non-culturable state in LisCVs and does not grow. Altogether, though occurring in epithelial cells, the features we describe for eSLAPs are in agreement with compartments similar to SLAPs, and distinct from LisCVs. Our observations that even low LLO activity allows *Lm* residence and replication in vacuoles is consistent with the previous report by Birmingham *et al.* showing that reduced *hlyA* expression allowed slow replication in macrophage SLAPs [12]. However, the replication of *Lm* inside eSLAPs is significantly faster than the 8 hours of doubling time reported in SLAPs [12], perhaps due to a lower bactericidal capacity of the epithelial niche compared with phagocytes. Membrane permeation by LLO might also attenuate the bactericidal properties of eSLAPs, and/or allow nutrient uptake through the permeated membrane, thereby promoting bacterial replication.

## Conclusion

Together with LisCVs and SLAPs, eSLAPs enrich the palette of *Lm* intravacuolar lifestyles that can establish in various cells types. Apprehending the importance of eSLAPs in the context of *in vivo* infections prompts future investigation. Indeed, whilst intravacuolar lifestyles impose constraints on motility or nutrient uptake, these compartments might provide shelter from cytosolic surveillance mechanisms such as autophagy and RIG-I-dependent activation of type-I interferon signalling, or favour chronic forms of infections by dampening cell-to-cell spread within tissues. Conversely, delayed residence within vacuoles could promote recognition of *Listeria* pathogen-associated molecular patterns by endosomal Toll-like receptors and activate NF-κB-dependent inflammatory pathways. Prolonged exposure to the intravacuolar environment could also tune the expression of *Lm* virulence genes. Deciphering the extent to which these intravacuolar niches influence the balance between bacterial fitness and host defences becomes critical to better appreciate long-term relationships between *Lm* and its host.

## Materials and methods

### Bacterial strains, plasmids and culture conditions

The bacterial source strains used in this work were *Escherichia coli* NEB5α (New England Biolabs) for plasmid constructions, Rosetta(DE3)pLysS (Novagen) for recombinant protein production, the clinical isolate of *Listeria monocytogenes* LL195 (lineage I, ST1) [39] for most of the experiments involving *Lm*, and *Shigella flexneri* M90T [40] for experiments on *Sf* T3SS-

dependent secretion. *Lm* reference strain EGD-e (lineage 2, ST9) [41] (lineage II, ST9) was also used as a control that the observed eSLAPs were not specific to LL195. All strains were grown at 37˚C under shaking at 190 rpm in Luria Bertani (LB) medium for *E. coli*, in LB or tryptic soy broth (TSB) for *Sf*, in brain hear infusion (BHI) or *Listeria* synthetic medium (LSM) [42] for *Lm*. Whenever required, media were supplemented with antibiotics for plasmid selection (chloramphenicol, 35 μg/ml for *E. coli*; 20 μg/ml for *Sf*; 7 μg/ml for *Lm* or Ampicillin, 100 μg/ml), or Congo red (1 mg/ml) for activation of the *Sf* T3SS.

In order to favour the expression of transgenes, the DNA coding sequence for FAST, fused with a Myc-tag, was codon-optimized for *Lm* or *Sf* using the online Optimizer application (http://genomes.urv.es/OPTIMIZER/) in guided random mode (S1 Text). The optimized sequences were obtained as synthetic Gene Fragments (Eurofins genomics). The *Lm*-optimized sequence additionally contained the 5'-untranslated (5'-UTR) of the *hlyA* gene, and the sequence encoding the signal peptide (SP) of LLO in its N-terminal part.

For plasmid constructions in the pAD vector derived from the pPL2 backbone [43,44], the 5'-UTR_{hlyA}-SP_{hlyA}-*FAST-Myc* fusion was amplified with primers oAL543-4, the sequence of the *Lm hlyA* gene encoding LLO was amplified from the EGD-e genomic DNA with primers oAL549-50b, and the coding sequence for eGFP was amplified from pAD-*cGFP* (BUG2479) [44] with primers oAL543-7. The UTR_{hlyA}-SP-FAST-Myc amplicon was inserted instead of UTR_{hlyA}-eGFP into the *Eag*I-*Sal*I restriction sites of pAD-*cGFP*, thus generating pAD-SP-FAST, where *FAST* is under control of the P_{HYPER} constitutive promoter (Fig 1A). pAD-*FAST*, pAD-*eGFP*, pAD-*SP-eGFP*, pAD-*hlyA*, pAD-*hlyA-FAST* and pAD-*hlyA-eGFP*, all containing the 5'-UTR of *hlyA* and a Myc tag, were likewise generated by inserting the cognate DNA amplicons into the same restriction sites (Fig 1A). After cloning in *E. coli* NEB5α, these plasmids were integrated in the genome of *L. monocytogenes* strains LL195 at the tRNA^{Arg} locus by electroporation as previously described [43]. The pHpPL3-*mCherry* plasmid was introduced at the tRNA^{Arg} locus by conjugation [45].

For allelic replacement at the *hlyA* locus (S5A Fig), pMAD-*ΔhlyA*::*FAST* and pMAD-*ΔhlyA*::*eGFP* were created by amplifying three partially overlapping fragments by PCR: one thousand base pairs (bp) upstream (*plcA* gene) and downstream (*mpl* gene) of the *hlyA* open reading frame in the EGD-e genome were amplified, respectively, with oAL981-2 and oAL976-7, while the FAST-Myc or eGFP-Myc open reading frames were amplified from pAD-FAST with oAL983-75 and from pAD-eGFP with oAL987-75, respectively. These fragments were inserted into the pMAD vector [46], between the *Sal*I and *Bgl*II restriction sites by Gibson Assembly, using the NEBuilder HiFi DNA Assembly Cloning Kit (New England Bio-Labs). pMAD-*hlyA-FAST* (S5A Fig) containing the last 1000 bp of *hlyA* fused with the FAST sequence, a Myc tag and one thousand bp downstream of *hlyA* was likewise generated by inserting the cognate DNA amplicons into the same restriction sites in pMAD. Site-directed mutagenesis to obtain pMAD-*hlyA*_{W492A}-*FAST* was carried out by PCR using oAL1073-74 as primers. Allelic replacements of the *hlyA* open reading frame by these constructs in the genomes of *L. monocytogenes* strains LL195 and EGD-e were obtained as previously described [46]. For complementation purposes in haemolysis assays, a simple in-frame deletion mutant of the *hlyA* gene was also created using the pMAD backbone.

For *Sf* constructs, *ipaB* and *ospF* were amplified from M90T genomic DNA with primers oAL703-4 and 707–8 respectively, and the optimized FAST-Myc was amplified with primers oAL705-6. pSU2.1-*ospF-FAST* (Fig 1C) was obtained by inserting an oAL707-6 amplicon overlapping *ospF* and FAST-Myc, with a *Bam*HI restriction linker, in place of mCherry into the *Kpn*I-*Xba*I restriction sites of pSU2.1rp-*mCherry* [47]. pSU2.1-*ipaB-FAST* was generated by replacing *ospF* with *ipaB* (oAL703-4) at the *Kpn*I-*Bam*HI sites (Fig 1C). After cloning in *E. coli* NEB5α, these plasmids were introduced in *Sf* M90T by electroporation.

The complete lists of bacterial strains and oligonucleotides used in this work are supplied as S1 and S2 Tables, respectively.

## Analysis of protein contents in bacterial total extracts or culture media

Bacterial total extracts or culture supernatants were recovered from 1 ml of *Lm* strains grown to an $OD_{600nm}$ of 2.0 in BHI at 37˚C as previously described [48].

Total bacterial extracts of *Sf* were prepared by boiling for 2 × 10 min at 95˚C in 100 μl of Laemmli sample buffer (SB 1X) the bacterial pellets obtained by centrifugation of 1 ml of each strain grown to an $OD_{600nm}$ of 2.0 in TCS medium at 37˚C. For assessment of secretion leakage prior to T3SS induction, 2 ml of *Sf* culture supernatants were collected, precipitated with 16% trichloroacetic acid (TCA), and centrifuged for 30 min at $16,000 \times g$ at 4˚C. Protein pellets were washed twice in acetone before resuspension in 50 μl of SB 1X. For induction of secretion, *Sf* were resuspended in 0.6 ml phosphate buffered saline (PBS) containing 1 mg/ml of Congo red at a final $OD_{600nm}$ of 40, and incubated at 37˚C for 45 min. Bacteria were eliminated by centrifugation; 100 μl of supernatant were collected and mixed with 33 μl of SB 4X for SDS-PAGE separation. The remainder supernatant was TCA-precipitated and resuspended in 50 μl SB 1X.

10 μl of each sample were separated on 4–15% Mini-Protean TGX gels (Bio-Rad) by sodium dodecyl sulfate-polyacrylamide gel electrophoresis (SDS-PAGE), and then revealed by staining with colloidal Coomassie Brilliant blue G-250 or by immunoblotting. For immunoblots, after transfer on nitrocellulose membrane (Amersham) using PierceG2 Fast Blotter, proteins were probed with anti-Myc mouse monoclonal antibody #9E10 (sc-40, Santa Cruz Biotechnology) at a 1:400 dilution in PBS supplemented with 0.05% tween-20 and 5% skimmed milk powder, followed by secondary hybridization with anti-Mouse IgG-heavy and light chain Antibody (Bethyl) at a 1:50 000 dilution in the same buffer. Signals were detected using Pierce ECL Plus Western Blotting Substrate and a Las4000 imager (GE Healthcare).

## Fluorescence measurements on culture supernatants

*Lm* were grown for 16 h in BHI, washed and diluted to 1:10 in *Listeria* synthetic medium (LSM), and then grown for 6 h at 37˚C, 180 rpm. Likewise, for secretion by *Sf* a culture in TSB was diluted to 1:10 in M9 medium supplemented with 0.2% glucose and 10 μg/ml nicotinic acid. From 1 ml of culture, bacterial pellets were collected by centrifugation of the cultures at $6,000 \times g$, then washed in PBS and resuspended in 1 ml of PBS. The culture supernatants were filtered (0.2 μm pores). For fluorescence measurements of FAST-tagged fusions, 180 μl of each sample was mixed with 20 μl of PBS containing 50 μM HBR-3,5DM ((Z)-5-(4-Hydroxy-3,5-dimethylbenzylidene)-2-thioxothiazolidin-4-one) to obtain a final concentration of 5 μM of fluorogen. Fluorescence intensity of technical triplicates was measured on a Spark 10M multimode microplate reader (Tecan), with excitation/emission wavelength set to 499/562 nm for FAST:HBR-3,5DM; 488/507nm for eGFP. Background fluorescence was measured on culture media from negative control strains (*Lm* LL195 [*pAD-LLO*] or *Sf* M90T Δ*ipaD*) and subtracted to each sample. The standard curve linking FAST fluorescence to its concentration was performed by diluting, in control medium corresponding to a culture of the corresponding negative control strain, known amounts of recombinant FAST produced in *E. coli* Rosetta(DE3)pLysS as previously described [17]. This enabled the calculation of fluorescent FAST-tagged proteins released in each culture.

As a negative control that the fluorescence was due to the formation of the FAST-HBR-3,5DM complexes, the fluorescence was also measured on samples mixed with 20 μl of PBS instead of PBS containing HBR-3,5DM. No fluorescence was detected above that of the control culture media.

For normalisation between measurements for FAST- and eGFP-tagged proteins, fluorescence intensities measured in filtered culture media were expressed relatively to the fluorescence measured for a suspension of $OD_{600nm}$ = 1 of *Lm* constitutively expressing either non-secreted FAST, or eGFP. The fluorescence intensities emitted by each one of these reference suspensions were fixed arbitrarily to 100 A.U.

Each experiment was reproduced three times independently. Statistical significance was assessed by two-tailed Student's *t*-tests with equal variance assumption, on the results of three independent experiments.

## Haemolysis assay

The supernatants of 16-h cultures of *Lm* in BHI were recovered by centrifugation for 1 min at $6,000 \times g$ followed by filtration through 0.2-μm pore filters, in order to eliminate bacteria. Serial two-fold dilutions of these supernatants were performed in round-bottom, clear, 96-well plates (100 μl final volume per well) using as a diluent PBS, the pH of which was adjusted to 5.6, and supplemented with 0.1% bovine serum albumin (BSA). Erythrocytes from defibrinated mice blood were washed twice in PBS pH 6.4 and diluted 1:10 in PBS pH 5.6. 50 μl of this suspension was added to each one of the wells containing diluted culture supernatants. After 30 min of incubation at 37°C, the plates were centrifuged for 10 min at $430 \times g$ and haemolytic titres were calculated as the reciprocal of the dilution for which 50% of haemolysis was observed [49]. Two-way ANOVA on $log_2$-transformed haemolytic titres followed by post-hoc Tukey test was used for statistical testing between conditions.

## Infection and transfection of epithelial cells

Infections of intestinal epithelial cells were performed in the LoVo cell line originating from colon adenocarcinoma (ATCC Cat# CCL-229, RRID: CVCL_0399). The Caco-2 epithelial cell line (ATCC Cat# HTB-37, RRID: CVCL_0025), also from colon adenocarcinoma, was used as a control that eSLAPs were not a specificity of LoVo cells. All cells were cultured in Dulbecco's Modified Eagle's medium (D-MEM) supplemented with 10% FBS, at 37°C in a humidified atmosphere containing 5% $CO_2$. For live microscopy, cells were seeded on Ibidi μslides 72 h prior to infection at a density of $10^5$ cells/ml, in 300 ml of culture medium. When needed, cells were transfected 24 h before infection with pEYFP-C1-CBD expressing YFP-CBD [14], using Lipofectamine LTX (Invitrogen) and 1 μg/ml of plasmid, according to the manufacturer's specifications.

*Lm* strains were grown in BHI until they reached early stationary phase ($OD_{600}$ of 2 to 3), washed in pre-warmed D-MEM, and then diluted in culture medium without serum to achieve a multiplicity of infection (MOI) of 5 (for long-term infections) to 30 (for short-term infections). Except for short-term imaging when bacterial entry was monitored under the microscope, after 1 h of bacteria-cell contact the inoculum was washed away twice with serum-free medium containing 40 μg/ml gentamicin, then the medium was replaced by complete culture medium without phenol red containing 25 μg/ml gentamicin in order to kill extracellular bacteria.

## Live fluorescence microscopy of infected cells

Infected cells were observed in D-MEM without phenol red supplemented with 5 μM of HBR-3,5DM for fluorescence detection, 250 nM of the fluorogenic probe SiR-actin for actin detection, and 25 μg/ml of gentamicin for long-term infections. For experiments where early events were monitored, the labelling of actin by SiR-actin was initiated 2 h prior to infection by adding 250 nM of SiR-actin to the medium.

For live cell imaging, preparations were observed with a Nikon Ti PFS microscope coupled to a spinning disk confocal device (CSU-XI-A1, Yokogawa), connected to a cooled EM-CCD camera (Evolve, Photometrics), and equipped with a cube for temperature control and a brick gas mixed for $CO_2$ and humidity control (Life Imaging Services). Image acquisition and microscope control were actuated with the MetaMorph software (Molecular Devices, RRID: SCR_002368). Fluorescent illumination was driven by three lasers, of wavelength 491 nm for eGFP, YFP or FAST, 561 nm for mCherry, and 635 nm for SiR-actin. Images were acquired with apochromat 63x objective lenses (NA 1.4) in 1 μm step-z-stacks. Acquisition parameters were similar for all samples of an experiment. For snapshot display, maximum intensity signals from 16 successive stacks (*i.e.* 16 μm slices) were integrated with Fiji (RRID:SCR_002285). Each picture or video is representative of the population observed in three independent experiments.

## Quantification of FAST accumulation in infected cells

For measurements of the accumulation of secreted FAST in cells, images were first z-projected by maximum intensity. The accumulation of fluorescence in a given cell was measured as the mean value of pixel intensities in a Region Of Interest (ROI) of 30 by 30 pixels. Statistical significance of the difference between conditions in the dispersion of fluorescence intensities over time was assessed by Two-tailed Mann-Whitney non-parametric test. The dynamics of FAST accumulation $I(t)$ was fitted to an exponential curve $I(t) = I_0 e^{rt} + I_{bg}$, with $r$ the rate of accumulation, $I_0$ the initial fluorescence and $I_{bg}$ the fluorescence of the background. The doubling time $\tau$ was then computed according to the following formula: $\tau = \frac{ln2}{r}$. Image computing was done with Fiji.

## *In-situ* growth measurements of mCherry-labelled bacteria

We performed for each time point an Otsu-thresholding on the entire z-stack of mCherry images in order to obtain the 3D segmentation of bacteria. Although the segmentation was not able to isolate individual bacteria in dense regions, it yielded the total volume occupied by bacteria in the field of view for each frame. The growth rate of mCherry bacteria was measured by fitting the total volume of segmented objects $V(t)$ to an exponential curve $V(t) = V_0 e^{rt}$, with $r$ the growth rate and $V_0$ the initial volume. The doubling time $\tau$ was then computed according to the following formula: $\tau = \frac{ln2}{r}$. Image computing was performed using MatLab (RRID: SCR_001622).

## Tracking of primary vacuoles in short term infection assays

The slices of the z-stacks obtained from spinning confocal imaging were projected onto a single plane (maximal projection). Fluorescent vacuoles were tracked using the plugin TrackMate in Fiji. The time at which tracks began during the infection was used to compute the time of *Lm* entry into LoVo cells. The distribution of residence times in primary vacuoles was computed from the statistics of track lengths. Statistical significance was assessed by two-tailed Student's *t*-test on the two distributions, with equal variance assumption. The interpolated median (*IM*) lifetime of SP-FAST-labelled vacuoles was calculated from the raw distributions as follows: $IM = M + \frac{(n_g - n_l)}{(2 \times n_e)} dt$, where $M$ is the median, $dt$ is the time interval, and $n_g$, $n_l$ and $n_e$ are the number of events greater, lower or equal to $M$, respectively. The accuracy of the interpolated median estimate was assessed by bootstrapping analysis (100 random samplings for each distribution, yielding 100 estimates of *IM* for each condition) followed by two-tailed Student's *t*-test on the distributions of the interpolated medians ($p << 10^{-15}$).

## Tracking of eSLAPs in long-term infection assays

At 2 h p.i., Ibidi μslides were mounted on a confocal spinning disk microscope for time-lapse observations. Pixel size was 250 x 250 nm and step size in *z* was 1 μm. For each time point taken every 5 or 15 minutes, we recorded a z-stack of fluorescent images in two channels for FAST signals and mCherry labelled bacteria. Given the good signal-to-noise ratio of mCherry images, we applied for each time point Otsu's thresholding algorithm on the entire z-stack in order to obtain a 3D segmentation of bacteria. The Otsu segmentation did not allow to isolate bacteria when they were too dense, for instance in eSLAPs, or after prolonged infection when cells were full of bacteria. Hence, we detected objects that could either be single bacteria or clusters of bacteria. We tracked each segmented object from frame to frame based on their similarities in size and location. Our method took benefit from the fact that cytosolic bacteria were moving very fast compared to those encapsulated in eSLAPs. The growth rates of bacteria inside eSLAPs were computed by fitting the dynamics of segmented mCherry volumes to an exponential function $V(t) = V_0 e^{rt}$, with *r* the growth rate and $V_0$ the initial volume of the vacuole. The doubling time *τ* was then computed according to the following formula: $\tau = \frac{ln2}{r}$. No growth was observed for the Δ*hlyA* strain in long-term vacuoles tracked by the co-occurrence of FAST and mCherry signals (based on the spherical shape of the object). For the WT and the *prfA\** strains, an object was classified as a growing eSLAP if it met the two following criteria: (*a*) the initial size was equal to or larger than the size a of single bacterium (32 voxels) (S12 Fig), and (*b*) the size of the object at least doubled over the whole track. The fraction of the objects in which *Lm* grew was then computed as the ratio of the number of tracked vacuoles that at least doubled their size during the 8-h course of the movie (thus matching *a* and *b*) to the initial number of objects bigger than 32 voxels on the first frame of the movie (S12 Fig), which is a good proxy for the initial number of entry events. To quantify LLO-FAST signals in a given vacuole, we used the binary mask of mCherry labelled objects to measure the average intensity in the corresponding region of the FAST image. Image computing was performed using MatLab.

## Quantification of YFP-CBD signals

Images were first z-projected by maximum intensity. For eSLAPs we measured YFP fluorescence intensity per pixel in ROIs with an area of 16 pixels that were manually positioned at each time point according to the mCherry signal of bacteria. For cytosolic bacteria, the mCherry images were segmented to get the masks of the individual bacteria. Segmented objects were scored as cytosolic if their size was below 40 pixels. At each time point, we measured the average value of fluorescence intensity per pixel in the population of bacteria that were cytosolic. The average background YFP-CBD signals per pixel measured in the cytosol was subtracted to each value for normalization.

## Immunofluorescence or LysoTracker staining of infected cells

LoVo cells were seeded 48 h before infection in 24-well plates containing 12 mm diameter coverslips. Infection with bacteria expressing mCherry (for immunofluorescence experiments) or eGFP (for LysoTracker staining) was performed as described above, using a MOI of 30, except that plates were centrifuged for 1 min at $200 \times g$ after addition of the inoculum in order to synchronise bacteria-cell contacts. 3 h p.i., cells were washed in pre-warmed PBS, fixed 20 min with 4% paraformaldehyde in PBS, then permeabilized for 5 min at room temperature with 0.5% Triton X-100 in PBS, and blocked for 5 min in PBS buffer containing 2% bovine serum albumin (BSA, Sigma). Incubation with primary antibodies in PBS buffer, 1% BSA was performed for 1 h, followed by three PBS washes, and incubation with the Alexa Fluor

647-conjugated secondary anti-rabbit antibody (Molecular probes Cat# A21245, RRID: AB_141775, 2 μg/μl), Acti-stain 488 phalloidin (Cytoskeleton #PHDG1, 70 nM) and DAPI (0.1 μg/μl) for 30 min. After three additional washes, cover glasses were finally mounted on microscope slides with Fluoromount mounting medium (Interchim). Rabbit monoclonal primary antibodies from Cell Signalling Technologies against Rab5 (Cat #3547, RRID: AB_2300649), Rab7 (Cat# 9367, RRID:AB_1904103) and LAMP1 (Cat# 9367, RRID: AB_2687579) were used at a 1:200 dilution; rabbit polyclonal antibodies against LC3 (MBL International Cat# PM036, RRID:AB_2274121) were used at a 1:500 dilution.

Staining of acidic compartments was obtained by adding 50 nM of LysoTracker Deed Red (Molecular Probes #L12492) to the cell culture medium 1 h prior to observation. Infected cells were then observed in DMEM without phenol red, supplemented with 500 ng/ml Hoechst 33342 and 25 μg/ml gentamicin.

Preparations were observed with a Nikon Ti epifluorescence microscope (Nikon), connected to a digital CMOS camera (Orca Flash 4.0, Hamamatsu). Illumination was achieved using a SOLA-SE 365 source (Lumencor) and the following excitation/emission/dichroic filter sets (Semrock): DAPI or Hoechst, 377(50)/447(60)/FF409-Di03; Acti-stain 488 phalloidin or eGFP, 472(30)/520(35)/FF495-Di03; mCherry, 562(40)/632(22)/dic FF562-Di03; Alexa 647 or LysoTracker, 630(30)/684(24)/dic FF655-Di01. Images were acquired with Nikon apochromat 60x objective lenses (NA 1.4). Image acquisition and microscope control were actuated with the μManager software (RRID:SCR_016865), and processed with Fiji. Each picture is representative of the infected cell population.

To quantify the number of vacuoles that associated with the indicated markers, 9 to 14 microscopic fields were examined from coverslips (or from live infection for the LysoTracker staining), and processed with Fiji. A vacuole was defined as a round aggregate of at least 4 bacteria (labelled with either mCherry in immunofluorescence experiments, or with eGFP in live experiments) that did not colocalize with actin staining. The presence of each marker was assessed by seeking for a corresponding fluorescent signal in the vicinity of vacuoles, and displaying a shape similar to the edge of the vacuole.

## Quantification of eGFP signals as a reporter for $P_{hlyA}$ induction in intracellular bacteria

LoVo cells were infected as described above for immunofluorescence experiments with bacteria that constitutively expressed mCherry, as well as eGFP under the control of the $P_{hlyA}$ promoter. At 1 and 3 h p.i., cells were fixed and stained with Acti-stain 670 phalloidin (Cytoskeleton #PHDN1-A, 70 nM) and DAPI as for immunofluorescence experiments. Fifteen to twenty fields per condition and time point were imaged with a Nikon Ti epifluorescence microscope as described above, and processed with Fiji. Images were first z-projected by maximum intensity, then bacteria were segmented using Otsu's thresholding algorithm on the mCherry signal. The objects were automatically counted and mapped on the images using the particle analyser. To split grouped bacteria, images were processed with Watershed segmentation algorithm and both mCherry and GFP average signals were measured. For each segmented bacterium, intrabacterial eGFP and mCherry signals were quantified, and intensity of the eGFP reporter of $P_{hlyA}$ activity was normalized to mCherry intensities. Kruskal-Wallis non-parametric test followed by Dunn's correction for multiple comparisons was used for statistical testing between conditions.

## Supporting information

**S1 Fig. Production and secretion of the Myc-tagged fusion proteins by *Listeria monocytogenes* LL195.** Protein production and secretion of Myc-tagged fusion proteins for each one of

the constructs described in Fig 1A (constitutive expression from an integrated pAD vector) was assessed by colloidal Coomassie staining (A, C) and immunoblotting with anti-Myc antibodies (B, D) of bacterial total extracts (A, B) and culture supernatant fractions (C, D) from 16-h cultures in BHI, separated by SDS-PAGE. (E) Epifluorescence microscopy observation of strains producing non-secreted FAST or eGFP. Scale bar, 2 μm. Most Myc-tagged protein constructs were detected by immunoblotting in the corresponding bacterial pellet fraction, indicating that transgenes were expressed, even though in varying amounts (B, lanes 2, 4–7). Constructs harbouring the LLO SP or full-length LLO were recovered in bacterial supernatants (C, D, lanes 3–7), suggesting that the SP of LLO promoted Sec-dependent export of not only of FAST or FAST-tagged proteins, but also of eGFP-fusion proteins. The secretion of eGFP-tagged proteins seemed less efficient than that of FAST-tagged protein (C, compare lane 3 with 4; D, compare lane 5 with 6), consistent with previous reports that eGFP is a poor substrate for Sec-dependent secretion[18]. Constructs devoid of signal peptides were not detected in supernatant fractions (C, D, lanes 1–2), arguing against the release of proteins into the culture medium due to bacterial lysis. For technical reasons likely due to the small size of FAST-Myc (15 kDa), it was not or barely detected by immunoblotting (B, D, lanes 1, 3); nevertheless, a strong signal corresponding to this polypeptide was visible on Coomassie-stained gels of the supernatant fractions, attesting of its secretion (C, lane 3). For bacterial pellet fractions (A, lanes 1, 3), signal from other proteins masked possible bands from that polypeptide; however, observation in microscopy (E) confirmed the non-secreted form of FAST was also produced. (TIF)

**S2 Fig. Fluorescent measurements of FAST- or eGFP-tagged proteins released in bacterial culture supernatants.** Six *Lm* strains expressing FAST- or eGFP-tagged proteins were cultured in LSM, then fluorescence intensities were measured on the filtered supernatants of each culture in presence of 5 μM HBR-3,5DM. For normalisation between FAST and eGFP signals, intensities were expressed in arbitrary units where 100 A.U. corresponds, for each reporter, to the intrabacterial fluorescence emitted by a suspension of equal volume of *Lm* ($OD_{600nm}$ = 1) that expresses constitutively either non-secreted FAST or eGFP under the $P_{HYPER}$ promoter. Residual fluorescence measured in the culture medium of strains producing non-secreted FAST or eGFP represents bacterial lysis. All values below 1 were considered below the detection limit for this experiment, and plotted as 1 (*i.e.* $10^0$). Normalized values, means and standard deviations from three independent experiments were plotted. *p*-values represent the results of two-tailed Student's *t*-tests with equal variance assumption. Source data are provided in S3 Table.
(TIF)

**S3 Fig. Production and secretion of the Myc-tagged fusion proteins by *Shigella flexneri* M90T.** Protein production and secretion of Myc-tagged fusion proteins for each one of the constructs described in Fig 1C (constitutive expression from a pSU2.1rp vector) was assessed by immunoblotting with anti-Myc antibodies of bacterial total extracts culture supernatant fractions, with or without induction of secretion by the T3SS using Congo red, and with or without TCA precipitation in order to concentrate samples. (A) Samples from wild type M90T *Sf*. (B) Samples from M90T Δ*ipaD*, in which T3SS secretion is constitutive. *, non-specific band.
(TIF)

**S4 Fig. Exponential growth of mCherry-labelled bacteria in infected LoVo cells.** Dynamics of the total intracellular bacterial population were measured by segmentation of mCherry-labelled bacteria, in control wells recorded in parallel to the accumulation of SP-FAST in the

cytoplasm (Fig 2). To get an estimate of the number of bacteria in each field, the total volume occupied by bacteria (the number of voxels that were labelled with mCherry) was divided by the average size of bacteria (32 voxels). Each colour represents an independent biological replicate (in blue, two wells were recorded in the same experiment). The exponential fit associated to each growth curve is displayed as orange dashed lines.
(TIF)

**S5 Fig. Construction, growth and haemolytic properties of the main *Lm* strains used in this study.** (A) Diagram of allelic replacement of *hlyA* (encoding LLO) at its chromosomal locus by a cassette expressing *SP-FAST* under the endogenous *hlyA* promoter (Δ*hlyA*:: *SP-FAST*), and of in-frame C-terminal tagging of LLO with FAST (*hlyA-FAST*). (B) Growth curves of three *Lm* LL195 strains harbouring pPL2-derived vectors, at 37˚C in BHI. No differences in growth rates were detected, regardless of the pPL2-derived plasmid that was integrated at the $tRNA^{Arg}$ locus (pAD-*SP-FAST* or pHpPL3-*mCherry*) and of the genetic modification carried out at the *hlyA* locus (Δ*hlyA* or Δ*hlyA*::*SP-FAST*). Curves represent the average and standard deviation of technical triplicates, displayed in linear scale (top) or in semi-log scale (bottom). (C) Haemolytic properties of the *Lm* strains producing FAST–or eGFP–tagged LLO fusions used in this study. The haemolytic titre measured for the strain where LLO was C-terminally tagged with FAST-Myc at the *hlyA* locus (*hlyA-FAST*) did not differ from that of the WT *Lm* strain. The haemolytic titre of all Δ*hlyA* strains was null (here, only Δ*hlyA*::*SP-FAST* was plotted). Haemolytic titres were enhanced for Δ*hlyA* deletion strains that had been complemented by integrative pAD plasmids harbouring *hlyA* fusion genes under control of the constitutive $P_{HYPER}$ promoter. Fusion with FAST-Myc or eGFP-Myc (pAD-*hlyA-FAST* or *-eGFP*) did not affect haemolytic properties, compared to a simple fusion with Myc (pAD-*hlyA*). None of these strains reached the intense haemolytic properties of the *prfA** strain (48.9-fold above the WT strain), for which the expression of *Lm* virulence genes (including *hlyA*) is deregulated, due to the constitutive activity of the transcriptional activator PrfA [22]. The average haemolytic titres and standard deviations from three independent experiments were plotted. Two-way ANOVA on $\log_2$-transformed haemolytic titres followed by post-hoc Tukey's test was used for statistical testing between conditions. ns, non-significant; ***, $p < 10^{-3}$, ****, $p < 10^{-4}$. Source data are provided in S7 Table.
(TIF)

**S6 Fig. Proliferation of *Lm* inside vacuoles when using the EGD-e strain in LoVo cells, or the LL195 strain in Caco-2 cells.** (A) LoVo cells infected with *Lm* EGD-e expressing both mCherry and LLO-FAST were observed between 2 and 8 h post-infection by spinning disk confocal microscopy. On the merged image, LLO-FAST is in cyan, mCherry is in orange, and SiR-actin is in purple. (B) Time-course of replication of mCherry-expressing *Lm* LL195 inside a vacuole, observed in the Caco-2 cell line. (A, B) Scale bars, 5 μm; timescale, h:min.
(TIF)

**S7 Fig. Quantitative and correlative analysis of the growth of bacteria and the secretion of LLO-FAST in eSLAPs over time.** LoVo cells were infected with *Lm* carrying an integrated pHpPL3-*mCherry* plasmid and secreting FAST-LLO due to an in-frame C-terminal fusion with FAST at the *hlyA* locus. (A) Number of bacteria inside eSLAPs over time. mCherry signals allowed the segmentation of bacteria and their counting. The exponential fit associated to each growth curve is displayed as orange dashed lines. The black horizontal dashed line represents the volume of one average doubling event since the first frame. (B) Quantification of the fluorescence over time in the FAST channel, which reports for the concentration of LLO-FAST in eSLAPs. (C-F) Correlation between the fluorescence generated by LLO-FAST in eSLAPs and

either the time residence time or the growth rate in these compartments. The average intensity of fluorescence generated by the secretion of LLO-FAST (C, E) and the maximum intensity of LLO-FAST fluorescence (D, F) were extracted for each eSLAP (n = 21) and correlated with the duration of this compartment since the beginning of acquisition (C, D) or with the growth rate of bacteria in this compartment, defined by the rate of increase of the size of the mCherry-labelled volume occupied by intravacuolar bacteria (E, F).
(TIF)

**S8 Fig. Growth of *Lm hlyA*-FAST [pHpPL3-*mCherry*] in infected LoVo cells.** The number of mCherry-labelled bacteria was determined by segmenting the volume they occupied, as in S4 Fig. Each colour represents an independent biological replicate. Curves of the same colour represent technical replicates. The exponential fit (linear fit in semi-log scale) associated to each growth curve is displayed as orange dashed lines.
(TIF)

**S9 Fig. Time-course of replication of mCherry-expressing *Lm* inside eSLAPs for WT, *prfA**\* *ΔhlyA* and *hlyA*~*W492A*~ *Lm* during the infection of LoVo cells.** The fluorescent signal of mCherry expressed constitutively was used to locate bacteria. Bacteria that were confined in eSLAPs were packed in a spherical configuration and observed as large spots on fluorescence images. When the vacuole ruptured, membrane tension was released and bacteria dispersed into the cytoplasm. Scale bars, 5 μm; timescale, h:min.
(TIF)

**S10 Fig. Expression of eGFP driven by the $P_{hlyA}$ promoter in intracellular WT or *prfA**\* bacteria.** LoVo cells infected with either WT or *prfA**\* EGD-e *Lm* strains, where eGFP was in transcriptional fusion with the promoter of *hlyA* by chromosomal allelic replacement (*ΔhlyA*:: *eGFP*), and co-expressing mCherry. Data represent the ratio of eGFP to mCherry signals for each segmented bacterium. The number of analysed bacteria per condition was n = 234 for WT and n = 846 for *prfA**\* bacteria at 1 h p.i.; n = 289 for WT and n = 2,124 for *prfA**\* bacteria at 3 h p.i. Means and standard deviations are represented in black solid lines. Note that the distribution of intensities was bimodal for the WT strain at 3 h p.i., likely reflecting the induction of $P_{hlyA}$ in some, but not all of the bacteria at this stage. *p*-values indicate the results of Kruskal-Wallis non-parametric test followed by Dunn's correction for multiple testing. Source data are provided in S8 Table.
(TIF)

**S11 Fig. Quantification of the co-localisation of eSLAPs with YFP-CBD and with markers of endosome maturation.** (A) YFP fluorescence intensity (reporting for the exposure of *Lm* to the host cytoplasm) was measured in bacteria before (dotted lines) or after (solid line) their release from eSLAPs, in the cell shown in Fig 5A and S5 Movie. The average background signal from the cytosol was subtracted and is represented as a black dashed line at 0 A.U. The orange trace displays the fate of a vacuole (dotted line) that ruptured at time 0 and released 11 bacteria (solid line) into the host cytosol. The blue trace represents a control vacuole in the same cell that did not rupture over the same time-course. Negative values correspond to signals below cytosolic levels, indicating that YFP-CBD was excluded from eSLAPs when they formed. In contrast, when bacteria were exposed to the host cytosol, they became positively stained. Error bars indicate the standard error of the mean. (B) The co-localisation of Rab5, Rab7, LC3 and LAMP1 with eSLAPs containing mCherry-labelled bacteria (immunofluorescence), or of LysoTracker Deep-red with GFP-labelled bacteria (live imaging), was assessed on infected LoVo cells at 3 h p.i. The occurrence of co-localisation events was 9% for Rab5 (2/22), 88,5% for Rab7 (23/26), 100% for LC3 (63/63), 80,5% for LAMP1 (33/41) and null for LysoTracker

(0/17). Source data are provided in S9 Table.
(TIF)

**S12 Fig. Distribution of object sizes on the first frames of time-lapses during the infection of LoVo cells by mCherry-labelled bacteria.** The first bin of the distribution (left of the green dotted line) corresponds to objects smaller than the size of bacteria that were discarded when counting the number of entry events. The bins between the green dotted and orange dashed lines correspond to single bacteria, the size of which was in the range of 32 to 64 voxels. The orange dashed line marks the limit between objects that correspond to individual bacteria (left) and clusters of bacteria (right). The mean of the distribution (n = 354 objects from 15 pooled experiments) was equal to 75 voxels.
(TIF)

**S1 Movie. Accumulation of secreted FAST in the cytoplasm of infected cells.** LoVo cells infected with *Lm* expressing SP-FAST were observed between 2 and 14 h post-infection by spinning disk microscopy. Scale bar, 10 μm.
(MOV)

**S2 Movie. Observation of secreted FAST signals in *Listeria* entry vacuoles.** LoVo cells infected with *Lm* expressing mCherry (A) or mCherry and SP-FAST (B) were observed between 0 and 3.25 h post-infection by spinning disk microscopy. Green, FAST channel (non-specific signals in A; secreted FAST in B); red, mCherry channel; blue, SiR-actin channel. Tracks for individual internalisation vacuoles containing mCherry-bacteria and SP-FAST are displayed in yellow. Scale bar, 10 μm.
(MOV)

**S3 Movie. Observation of the decoration of vacuoles by LLO-FAST in *Listeria* cells infected by *Lm* LL195.** LoVo cells infected with *Lm* LL195 expressing both mCherry and LLO-FAST were observed between 2 and 8 h post-infection by spinning disk microscopy. Orange, FAST channel; cyan, mCherry channel; purple, SiR-actin channel. Scale bar, 5 μm.
(MOV)

**S4 Movie. Observation of the decoration of vacuoles by LLO-FAST in *Listeria* cells infected by *Lm* EGD-e.** LoVo cells infected with *Lm* EGD-e expressing both mCherry and LLO-FAST were observed between 2 and 9 h post-infection by spinning disk microscopy. Orange, FAST channel; cyan, mCherry channel; purple, SiR-actin channel. Scale bar, 5 μm.
(MOV)

**S5 Movie. Imaging of the differential labelling by CBD-YFP of the cytosolic *versus* intravacuolar populations of intracellular bacteria.** LoVo cells were transfected with pEYFP-CBD 24 h being infected with *Lm* expressing mCherry, then imaged by spinning disk microscopy from 2 to 8 h p.i. Cyan, CBD-YFP channel; orange, mCherry channel. The cell outline is shown with a purple dashed line. Scale bar, 5 μm.
(MOV)

**S1 Text. Synthetic gene fragments with codon-optimized sequences for expression of FAST fusions in *Listeria* or *Shigella*.**
(PDF)

**S1 Table. Bacterial strains.**
(XLSX)

**S2 Table. Oligonucleotides.**
(XLSX)

**S3 Table. Source data for Fig 1 and S2 Fig.**
(XLSX)

**S4 Table. Source data for Fig 2.**
(XLSX)

**S5 Table. Source data for Fig 3.**
(XLSX)

**S6 Table. Source data for Fig 4.**
(XLSX)

**S7 Table. Source data for S5 Fig.**
(XLSX)

**S8 Table. Source data for S10 Fig.**
(XLSX)

**S9 Table. Source data for S11 Fig.**
(XLSX)

# Acknowledgments

We are grateful to Marie-Aude Plamont, Vinko Besic, Sebastian Rupp and Alison Tebo for precious experimental assistance and eagerness to help solve technical issues. We thank Lionel Schiavolin and Didier Filopon for providing source strains and practical advice regarding *Sf* experiments, Jost Enninga and Hélène Bierne for insightful discussion. We thank the IBENS imaging facility for maintaining access to microscopy equipment that was instrumental to this work, and providing expert support whenever needed. We are indebted to the IBENS animal facility and especially to Eléonore Touzalin for kindly supplying the mice blood used in hae-molysis assays.

# Author Contributions

**Conceptualization:** Caroline Peron-Cane, Arnaud Gautier, Nicolas Desprat, Alice Lebreton.

**Data curation:** Caroline Peron-Cane, Nicolas Desprat, Alice Lebreton.

**Formal analysis:** Caroline Peron-Cane, Nicolas Desprat.

**Funding acquisition:** Alice Lebreton.

**Investigation:** Caroline Peron-Cane, José-Carlos Fernandez, Julien Leblanc, Laure Wingerts-mann, Nicolas Desprat, Alice Lebreton.

**Methodology:** Caroline Peron-Cane, Arnaud Gautier, Nicolas Desprat, Alice Lebreton.

**Project administration:** Caroline Peron-Cane, Nicolas Desprat, Alice Lebreton.

**Resources:** Arnaud Gautier, Nicolas Desprat, Alice Lebreton.

**Software:** Nicolas Desprat.

**Supervision:** Nicolas Desprat, Alice Lebreton.

**Validation:** Caroline Peron-Cane, José-Carlos Fernandez.

**Visualization:** Caroline Peron-Cane, Nicolas Desprat, Alice Lebreton.

**Writing – original draft:** Nicolas Desprat, Alice Lebreton.

**Writing – review & editing:** Caroline Peron-Cane, Arnaud Gautier, Nicolas Desprat, Alice Lebreton.

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
