## [Decision Letter · Decision Letter 0]

19 Jul 2020

Dear Dr. Lebreton,

Thank you very much for submitting your manuscript "Fluorescent secreted bacterial effectors reveal an intravacuolar replication compartment for Listeria monocytogenes" for consideration at PLOS Pathogens. As with all papers reviewed by the journal, your manuscript was reviewed by members of the editorial board and by several independent reviewers. The reviewers appreciated the attention to an important topic. Based on the reviews, we are likely to accept this manuscript for publication, providing that you modify the manuscript according to the review recommendations.

Sincerely,

Renée M. Tsolis

Section Editor

PLOS Pathogens

Renée Tsolis

Section Editor

PLOS Pathogens

Kasturi Haldar

Editor-in-Chief

PLOS Pathogens

orcid.org/0000-0001-5065-158X

Michael Malim

Editor-in-Chief

PLOS Pathogens

orcid.org/0000-0002-7699-2064

Reviewer Comments (if any, and for reference):

Reviewer's Responses to Questions

**Part I - Summary**

Reviewer #1: In this paper the authors developed a method to track secreted proteins in live microscopy. Authors show application of the new tracking system to two markedly different secretion systems in both Gram positive and Gram negative intracellular pathogens. They then used their tool to measure time of vacuole rupture in Listeria, which is an important step in Listeria infection cycle, freeing bacteria and allowing infection of the neighbouring cell. They observed that during the infection not all of the bacteria succeed in escaping the vacuole. This phenomenon, which occurs despite expression of the listeriolysin O (LLO) pore forming toxin, allows a subpopulation of Listeria to proliferate within these vacuoles reminiscent of macrophage SLAPs, from which they eventually escape. The eSLAP (for epithelial SLAP) therefore represents another replicative niche used by Listeria (and possibly other intracellular bacteria) to a colonize host cell. In addition, this piece of work provides an interesting tool that is relevant and potentially applicable to many bacterial organisms. The work should therefore be of interest for a large community of bacteriologists.

The paper is well written and easy to follow. Overall, this work is of outstanding quality and well supported by the results with appropriate controls and statistical analysis. I reviewed it earlier for another journal and the authors have addressed all the comments I had then. This is now a version that can be published without further delays.

Reviewer #2: This is an interesting work that describes the use of a FAST reporter to study the localization of LLO from Listeria in real time. Using a LLO-FAST reporter, the authors find that, in epithelial cell lines, a subpopulation of Listeria fails to escape from phagosomes and instead replicate in these structures in a manner dependent on LLO. They refer to phagosomes in epithelial cells that contain replicating Listeria as ‘eSLAPS’.

This is a well performed study that provides new insight into the functions of LLO. I provide a few comments and questions below.

**Part II – Major Issues: Key Experiments Required for Acceptance**

Reviewer #1: The paper is acceptable for publication as is, authors have taken on board all comments from a previous round of reviewing in which I happened to take part.

Reviewer #2: 1. I could not find any information on the method of statistical analysis used in this study. In my view, it is critical to mention the statistical test used in each figure so that one can judge if the statistical analysis is appropriate.

2. Does replication in eSLAPs require the pore-forming activity of LLO?

3. Line 251. Just because the PrfA* mutant form of PrfA is constitutively activated in broth culture does not necessarily mean it is more active than wild-type PrfA in host cells, given that PrfA is activated in cells. Are there published data demonstrating that a PrfA* mutant stimulates higher expression of PrfA-dependent genes in host cells compared to wild-type bacteria? Although the images in Fig. S9 give the general impression that LLO-FAST accumulates to a greater extent in eSLAPs from PrfA* bacteria compared to wild-type bacteria, this apparent difference is not quantified. I would suggest quantification from several eSLAPs and the use of statistical analysis to determine if PrfA* does, in fact, cause increased accumulation of LLO-FAST.

**Part III – Minor Issues: Editorial and Data Presentation Modifications**

Reviewer #1: None

Reviewer #2: 1. Figure 2C. It is difficult for this referee to distinguish between the green points and the blue curve. I would suggest changing blue to another color. Actually, black might work fairly well.

2. Line 191, ‘significantly longer’; line 193 ‘a significant proportion’.l I would avoid the use of ‘significant’ in these contexts, since statistical analysis did not seem to be performed on Fig. 3C or Fig. 3D. In a related matter, I wonder why statistical analysis was not carried out on Fig. 3D. It would seem to be important to determine if the difference between the WT and ∆hlyA strain is significant.

3. Fig. S7A. What do the dashed lines represent?

4. Fig. S7B. How does one account for the fluctuating in LLO-FAST signals in some of the vacuoles (Fig. S7B)? Is LLO-FAST being degraded and then re-synthesized?

5. Line 248. Should be Fig. 4C (not Fig. 3C).

6. Line 251. What is meant by ‘Symmetrically’?

7. Line 279 and Fig. 5A. To this referee, the evidence that eSLAP rupture precedes decoration of bacteria with YFP-CBD does not seem that compelling in this image. The CBD labeling seems quite weak and I’m not convinced that CBD-labeled bacteria appear in the same area of the cytosol that previously was occupied by the vacuolar structure labeled with Lm. Perhaps the authors could select a clearer set of images?

PLOS authors have the option to publish the peer review history of their article (what does this mean?). If published, this will include your full peer review and any attached files.

Reviewer #1: No

Reviewer #2: No
---

## [Editor Report · Decision Letter 1]

21 Sep 2020

Dear Dr. Lebreton,

We are pleased to inform you that your manuscript 'Fluorescent secreted bacterial effectors reveal active intravacuolar proliferation of Listeria monocytogenes in epithelial cells' has been provisionally accepted for publication in PLOS Pathogens.

Thank you for your detailed and thoughtful response to Reviewer #2. We look forward to publication of this exciting work!

Best regards,

Renée Tsolis

Section Editor

PLOS Pathogens

Kasturi Haldar

Editor-in-Chief

PLOS Pathogens

orcid.org/0000-0001-5065-158X

Michael Malim

Editor-in-Chief

PLOS Pathogens

orcid.org/0000-0002-7699-2064

---

## [Editor Report · Acceptance letter]

2 Oct 2020

Dear Dr. Lebreton,

We are delighted to inform you that your manuscript, "Fluorescent secreted bacterial effectors reveal active intravacuolar proliferation of <I>Listeria monocytogenes</I> in epithelial cells," has been formally accepted for publication in PLOS Pathogens.

Best regards,

Kasturi Haldar

Editor-in-Chief

PLOS Pathogens

orcid.org/0000-0001-5065-158X

Michael Malim

Editor-in-Chief

PLOS Pathogens

orcid.org/0000-0002-7699-2064